# Quantitative SUMO proteomics identifies PIAS1 substrates involved in cell migration and motility

Chongyang Li[1,2], Francis P. McManus[1], Cédric Plutoni[1], Cristina Mirela Pascariu[1], Trent Nelson[1,2], Lara Elis Alberici Delsin[1,2], Gregory Emery [1,3] & Pierre Thibault[1,4,5 ✉]

The protein inhibitor of activated STAT1 (PIAS1) is an E3 SUMO ligase that plays important roles in various cellular pathways. Increasing evidence shows that PIAS1 is overexpressed in various human malignancies, including prostate and lung cancers. Here we used quantitative SUMO proteomics to identify potential substrates of PIAS1 in a system-wide manner. We identified 983 SUMO sites on 544 proteins, of which 62 proteins were assigned as putative PIAS1 substrates. In particular, vimentin (VIM), a type III intermediate filament protein involved in cytoskeleton organization and cell motility, was SUMOylated by PIAS1 at Lys-439 and Lys-445 residues. VIM SUMOylation was necessary for its dynamic disassembly and cells expressing a non-SUMOylatable VIM mutant showed a reduced level of migration. Our approach not only enables the identification of E3 SUMO ligase substrates but also yields valuable biological insights into the unsuspected role of PIAS1 and VIM SUMOylation on cell motility.

[1] Institute for Research in Immunology and Cancer, Université de Montréal, Montréal, Québec, Canada. [2] Molecular Biology Program, Université de Montréal, Montréal, Canada. [3] Department of Pathology and Cell Biology, Université de Montréal, Montréal, Québec, Canada. [4] Department of Chemistry, Université de Montréal, Montréal, Québec, Canada. [5] Department of Biochemistry and Molecular Medicine, Université de Montréal, Montréal, Québec, Canada. ✉email: pierre.thibault@umontreal.ca

The small ubiquitin-like modifier (SUMO) protein is an ubiquitin-like (UBL) protein that is highly dynamic and can reversibly target lysine residues on a wide range of proteins involved in several essential cellular events, including protein translocation and degradation, mitotic chromosome segregation, DNA damage response, cell cycle progression, cell differentiation, and apoptosis[1]. SUMO proteins are highly conserved through evolution and the human genome encodes four SUMO genes, of which three genes (SUMO1, SUMO2, and SUMO3) are ubiquitously expressed in all cells[1,2]. Prior to conjugation, the immature SUMO proteins are C-terminally processed by sentrin-specific proteases[3]. These proteases also cleave the isopeptide bond formed between the ε-amino group of the acceptor lysine residues and the C-terminus residue of the conjugated SUMO proteins. The conjugation of SUMO to target proteins requires an E1-activating enzyme (SAE1/2), an E2-conjugating enzyme (UBC9), and one of several E3 SUMO ligases[4]. Unlike ubiquitination, in vitro SUMOylation can occur without E3 SUMO ligases, although enhanced substrate specificity is conferred by E3 SUMO ligases[5]. It is believed that SUMOylation events occurring without the aid of E3 SUMO ligases arise primarily on the consensus motif composed of ψKxE, where ψ represents a large hydrophobic residue and x represents any amino acid[6]. To date, several structurally unrelated classes of proteins appear to act as E3 SUMO ligases in mammalian cells, such as the protein inhibitor of activated STAT (PIAS) family of proteins, Ran-binding protein 2, the polycomb group protein (Pc2), and topoisomerase I- and p53-binding protein (TOPORS)[7,8].

PIAS orthologs can be found through eukaryote cells and comprise four PIAS proteins (PIAS1, PIASx (PIAS2), PIAS3, and PIASy (PIAS4)), which share a high degree of sequence homology[9]. Overall, five different domains or motifs on PIAS family proteins recognize distinct sequences or conformations on target proteins, unique DNA structures, or specific "bridging" molecules to mediate their various functions[10]. An example of this is the SAF-A/B, Acinus and PIAS (SAP) domain, which has a strong affinity towards A–T-rich DNA[11] and binds to Matrix attachment regions DNA[12], in addition to having an important role in substrate recognition[13]. The PINIT motif affects subcellular localization and contributes to substrate selectivity[14,15]. The Siz/PIAS RING (SP-RING) domain interacts with UBC9 and facilitates the transfer of SUMO to the substrate[16]. The PIAS SIM (SUMO interaction motif) recognizes SUMO moieties of modified substrates and alters subnuclear targeting and/or assembly of transcription complex[16–18]. Although several functions have been attributed to these domains, relatively little is known about the role of the poorly conserved C-terminus serine/threonine-rich region.

PIAS1 is one of the most well-studied E3 SUMO ligases and was initially reported as the inhibitor of signal transducers and activators of transcription 1 (STAT1)[19]. Previous studies indicated that PIAS1 interacts with activated STAT1 and suppresses its binding to DNA[8]. PIAS1 overexpression was reported in several cancers, including prostate cancer, multiple myeloma, and B-cell lymphomas[20–23]. PIAS1 can SUMOylate the focal adhesion kinase (FAK) at Lys-152, a modification that dramatically increases its ability to autophosphorylate Thr-397, activate FAK, and promote the recruitment of several enzymes including Src family kinases[24]. In yeast, Lys-164 SUMOylation on proliferating cell nuclear antigen (PCNA) is strictly dependent on the PIAS1 ortholog Siz1 and is recruited to the anti-recombinogenic helicase Srs2 during S-phase[25]. PIAS1 can also regulate oncogenic signaling through the SUMOylation of promyelocytic leukemia (PML) and its fusion product with the retinoic acid receptor-α (PML-RARα) as observed in acute PML (APL)[26]. In addition to its regulatory role in PML/PML-RARα oncogenic signaling,

PIAS1 has been shown to be involved in the cancer therapeutic mechanism of arsenic trioxide (ATO). This is accomplished by ATO promoting the hyperSUMOylation of PML-RARα in a PIAS1-dependent manner, resulting in the ubiquitin-dependent proteasomal degradation of PML-RARα and APL remission[26]. In B-cell lymphoma, PIAS1 has been reported as a mediator in lymphomagenesis through SUMOylation of MYC, a proto-oncogene transcription factor associated with several cancers. SUMOylation of MYC leads to a longer half-life and therefore an increase in oncogenic activity[23]. Altogether, these reports suggest that PIAS1 could promote cancer cell growth and progression by regulating the SUMOylation level on a pool of different substrates.

In this study, we first evaluate the effects of PIAS1 overexpression in HeLa cells. PIAS1 overexpression has a significant influence on cell proliferation, cell migration, and motility. To identify putative PIAS1 substrates, we develop a system-level approach based on quantitative SUMO proteomic analysis[27], to profile changes in protein SUMOylation in cells overexpressing this E3 SUMO ligase. Our findings reveal that 91 SUMO sites on 62 proteins were regulated by PIAS1. Bioinformatic analysis indicates that many PIAS1 substrates are involved in transcription regulation pathways and cytoskeleton organization. Interestingly, several PIAS1 substrates, including cytoskeletal proteins (actin filaments, intermediate filaments (IFs), and microtubules), are SUMOylated at lysine residues located in non-consensus motif. We confirm the SUMOylation of several PIAS1 substrates using both a reconstituted in vitro and cell-based in vitro SUMOylation assays. Further functional studies reveal that PIAS1 mediates the SUMOylation of vimentin (VIM) at two conserved sites on its C-terminus that affect the dynamic disassembly of this IF protein.

## Results

**PIAS1 regulates HeLa cell proliferation and motility**. To investigate the physiological function of PIAS1 in HeLa cells, we overexpressed PIAS1 (Fig. 1a) and generated a PIAS1-knockout (KO) cell line (Supplementary Fig. 1 and Fig. 1b). For the PIAS1 overexpression, the abundance of PIAS1 in HeLa cells was increased by six-fold at 48 h post transfection (Fig. 1a). PIAS1 overexpression promoted HeLa cell proliferation by ~50% (Fig. 1c), whereas the KO cells reduced the rate of proliferation by ~50% (Fig. 1d). We further examined the phenotypic effects of PIAS1 expression on cell migration using the wound-healing assay. The migration ability of Hela cells was increased after PIAS1 overexpression (Fig. 1e), whereas the KO cells displayed a reduced rate of migration (Fig. 1g). Taken together, these results highlight the role that PIAS1 plays in regulating cell growth and cell migration of HeLa cells.

**Identification of PIAS1 substrates by SUMO proteomics**. To gain a better understanding of the role that PIAS1 plays in cell proliferation, migration, and motility, we modified our previously published large-scale SUMO proteomic approach to identify PIAS1 substrates in a site-specific manner (Fig. 2)[28]. We combined a SUMO remnant immunoaffinity strategy[27] with metabolic labeling (stable isotope labeling of amino acid in cell culture, SILAC) to study the global changes in protein SUMOylation upon PIAS1 overexpression. HEK293 cells stably expressing SUMO3m (Supplementary Fig. 2) were grown at 37 °C in media containing light ($^0$Lys, $^0$Arg), medium ($^4$Lys, $^6$Arg), or heavy ($^8$Lys, $^{10}$Arg) isotopic forms of lysine and arginine. Three biological replicates were performed, and for each replicate one SILAC channel was transfected with an empty vector while the other two were transfected with Myc-PIAS1 vectors (Fig. 2a). At 48 h post

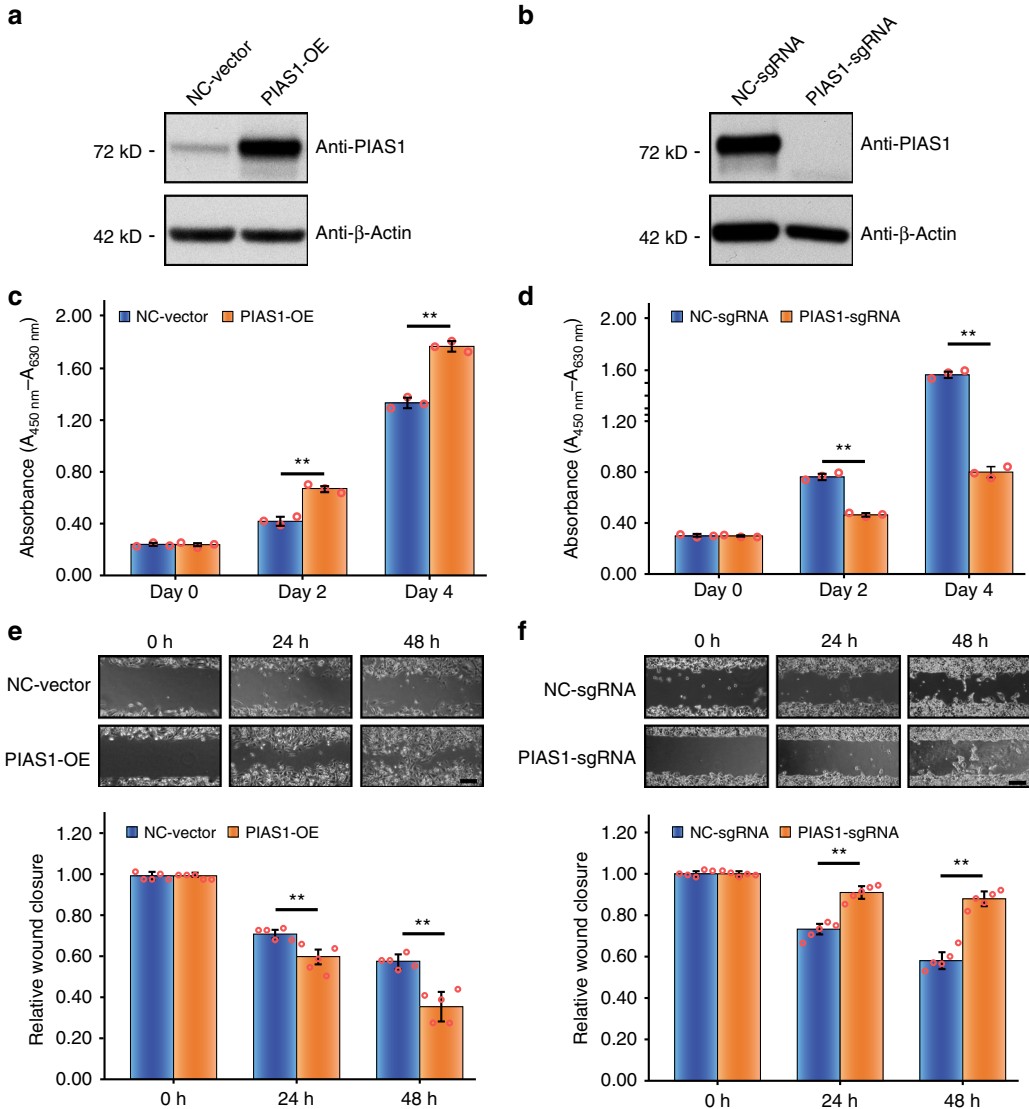

**Fig. 1 Functional effects of PIAS1 expression on HeLa cells. a** HeLa cells were transfected with Myc-PIAS1 (PIAS1-OE) or Empty vector (NC-vector) for 48 h. PIAS1 overexpression efficiency was determined by western blotting. Actin was used as a loading control. **b** PIAS1 KO HeLa cells were generated by CRISPR/Cas9. PIAS1 KO was determined by western blotting. Actin was used as a loading control. **c** PIAS1 overexpression in HeLa cells significantly promotes cell growth, $n = 3$ biologically independent samples. **d** PIAS1 KO HeLa cells (PIAS1-sgRNA) showed impeded cell growth compared with the negative control cells (NC-sgRNA), $n = 3$ biologically independent samples. **e** PIAS1 overexpression increased cell migration as determined by a wound-healing assay. Scale bar: 200 μm, $n = 5$ biologically independent samples. **f** PIAS1 KO cells displayed decreased cell migration compared with the sgRNA-negative control cells, as determined by a wound-healing assay. Scale bar: 200 μm, $n = 5$ biologically independent samples. Data represent the mean ± SD, error bars represent SD, **$p < 0.01$, Student's $t$-test. Source data are provided as a Source Data file.

transfection, an equal amount of cells from each SILAC channel were collected and combined before lysis in a highly denaturant buffer. PIAS1 overexpression efficiency in HEK293-SUMO3m cells was evaluated by western blotting (Fig. 2b). Protein extracts were first purified by NiNTA beads to enrich SUMO-modified proteins and digested on beads with trypsin (Fig. 2c). Following tryptic digestion, SUMO-modified peptides were immunopurified using an antibody directed against the NQTGG remnant that is revealed on the SUMOylated lysine residue. Next, peptides were fractionated by offline strong cation exchange (SCX) STAGE tips and analyzed by liquid chromatography–tandem mass spectrometry (LC-MS/MS) on a Tribrid Fusion instrument. To determine that abundance changes were attributed to SUMOylation and not to change in protein expression, we also performed quantitative proteomic analysis on the total cell extracts (TCEs) from PIAS1 overexpression (Fig. 3a and Supplementary Fig. 3).

PIAS1 overexpression caused a global increase in protein SUMOylation with negligible changes on protein abundance (Fig. 3b). In total, 12,080 peptides on 1756 proteins (Fig. 3a and Supplementary Data 1) and 983 SUMO peptides on 544 SUMO proteins (Fig. 3b and Supplementary Data 2) were quantified for the proteome and SUMO proteome analyses, respectively. A total of 91 SUMOylation sites on 62 proteins were found to be upregulated by PIAS1 overexpression including its known substrate PML protein. A summary of these analyses is shown in Fig. 3c.

Protein classification ontology analysis of the PIAS1 substrates using PANTHER clustered the targets into 11 groups (Fig. 3d). PIAS1-mediated SUMOylation predominantly occurred on nucleic acid-binding proteins, transcription factors, cytoskeletal proteins, chaperone proteins, enzyme modulators, and ligases. We next classified putative PIAS1 substrates by their Gene Ontology (GO) molecular function, biological process, and

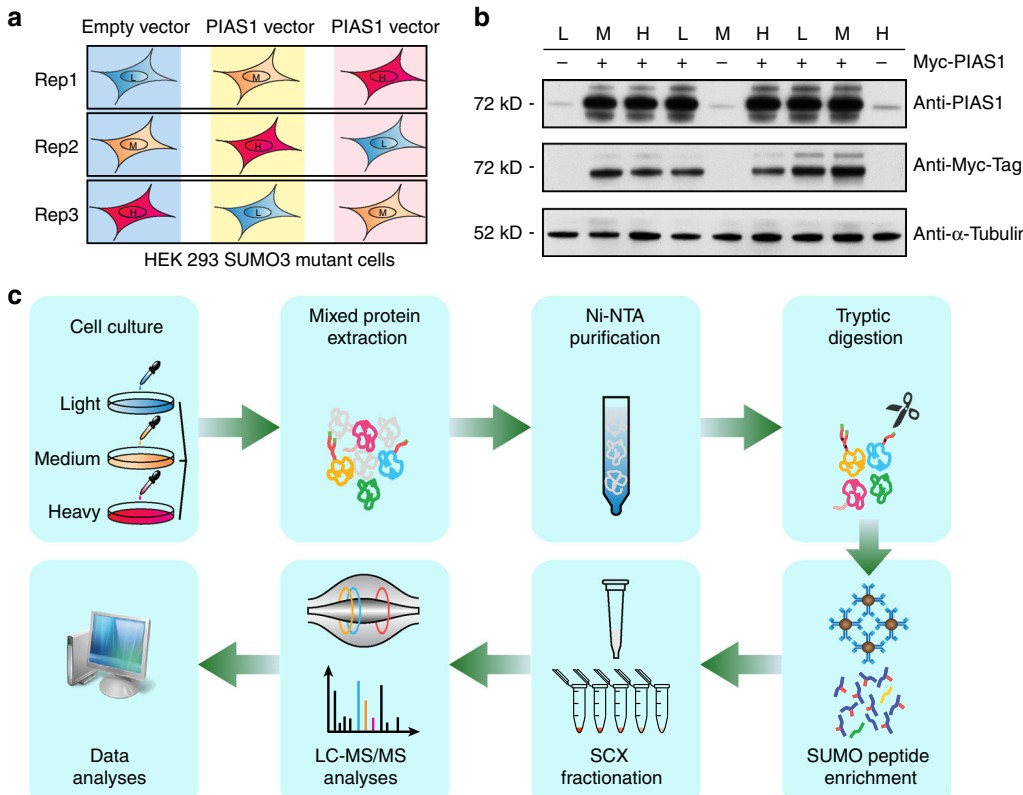

**Fig. 2 Workflow for the identification of PIAS1 substrates. a** HEK293-SUMO3m cells were cultured in SILAC medium with reverse labeling in biological triplicates. For replicate 1, PIAS1 overexpression was performed in medium and heavy channels with Myc-PIAS1 vector, whereas the cells cultured in light media were transfected with the pcDNA3.0-Myc vector (empty vector). For replicates 2 and 3, the empty vector was transduced in the medium and heavy labeled cells, respectively. **b** Western blotting showing the level of overexpression of PIAS1 in the transfected cells. Detection of PIAS1 overexpression by both Anti-PIAS1 antibody and Anti-Myc antibody. α-Tubulin was used as loading control. **c** SILAC-labeled cells were lysed and combined in a 1:1:1 ratio based on protein content. SUMOylated proteins were enriched from the cell extract on an IMAC column prior to their tryptic digestion. After desalting and drying, peptides containing the SUMO3 remnant were enriched using a custom anti-K-ε-NQTGG antibody that was crosslinked on magnetic beads. Enriched peptides were further fractionated on SCX columns and injected on a Tribrid Fusion mass spectrometer. Peptide identification and quantification were performed using MaxQuant. Source data are provided as a Source Data file.

cellular components (Fig. 3e) using the whole identified SUMOylome as the background. GO cellular component classification revealed that PIAS1 substrates were enriched in PML body, plasma membrane, and microtubule compared with the global SUMOylome (Fig. 3e). GO biological process analysis revealed that identified PIAS1 substrates are involved in a variety of biological processes, including protein stabilization, protein SUMOylation, and protein folding (Fig. 3e). GO molecular function analysis indicated that PIAS1 substrates are associated with ubiquitin protein ligase binding, structural molecule activity, protein tag, SUMO transferase activity, and unfolded protein binding (Fig. 3e). Indeed, PIAS1 regulates the SUMOylation of several proteins whose roles in the cell are diverse. Much like global SUMOylation, PIAS1-mediated SUMOylation may play a role in several biological processes that are independent from each other.

Previous SUMO proteome analyses indicated that under unstressed conditions, approximately half of acceptor lysine residues are found in the SUMO consensus and reverse consensus motifs[29]. As SUMOylation is believed to occur at UBC9 consensus site without an E3 SUMO ligase, we surmised that PIAS1-mediated SUMOylation may occur at non-consensus motifs. We therefore compared the amino acid residues surrounding the SUMOylation sites that are regulated by PIAS1 with those of the whole SUMO proteome (Supplementary Fig. 4a). As anticipated, the sequences surrounding the PIAS1-mediated

SUMOylation sites are depleted in glutamic acid at position +2 and depleted of large hydrophobic amino acids at position −1, consistent with the reduction of the consensus sequence. Indeed, E3 SUMO ligases appear to aid in the SUMOylation of lysine residues that reside in non-canonical regions. Furthermore, we investigated the local secondary structures and solvent accessibility of PIAS1 substrates surrounding SUMO sites using NetSurfP-1.1 software (Supplementary Fig. 4b). We observed that one-third of PIAS1-regulated SUMOylation sites are located within α-helix and approximately one-tenth within β-strand. In contrast, the majority of SUMOylated lysine residues in the SUMO proteome are localized in coil regions. Taken together, these results support the notion that PIAS1-mediated SUMOylation preferentially occurs on structured regions of the protein, which may help substrate recognition by PIAS1. In addition, we noted that PIAS1-mediated SUMOylation occurred primarily on solvent-exposed lysine residues, which was also the case for the global SUMOylome. These results suggest that PIAS1 may not impart conformational changes to its substrate upon binding, as it does not promote SUMOylation on lysine residues that would otherwise be buried within the core of the substrate. Overall, PIAS1 promotes the ability of UBC9 to SUMOylate lysine residues that are present in non-consensus sequences located on ordered structures of the proteins.

To better understand the cellular processes regulated by PIAS1, a Search Tool for the Retrieval of Interacting Genes/Proteins

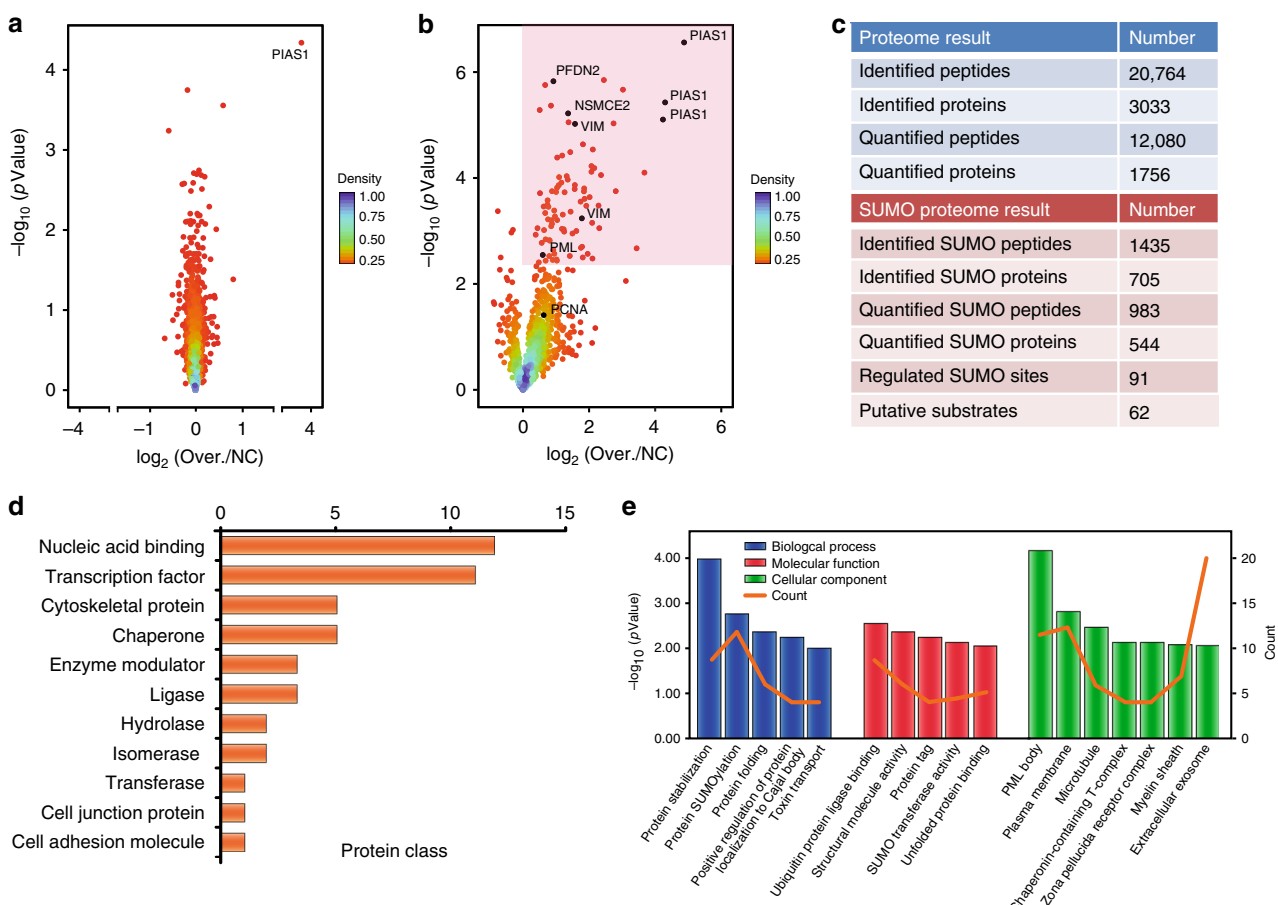

**Fig. 3 Mass spectrometry results and bioinformatic analyses of identified PIAS1 substrates. a** Volcano plot showing the global proteome changes in cells overexpressing PIAS1 (Over.) vs. control cells (NC). Individual proteins are represented by points. **b** Volcano plot showing the global SUMOylation changes in cells overexpressing PIAS1 (Over.) vs. control cells (NC). Individual SUMOylation sites are represented by points. The area of the volcano plot where SUMO sites are significantly upregulated (Benjamini–Hochberg corrected $p$-value of <0.05) is shaded in pink. **c** Summary of identified and quantified peptides and proteins in both the proteome and SUMOylome experiments. **d** Functional classification of PIAS1 substrates using PANTHER (Protein Analysis Through Evolutionary Relationships) (http://www.pantherdb.org). **e** GO term enrichment distribution of the identified PIAS1 substrates using DAVID 6.8 (https://david.ncifcrf.gov/). Source data are provided as a Source Data file.

(STRING) analysis was performed to analyze the interaction network of putative PIAS1 substrates. This network highlights the presence of highly connected interactors from PML nuclear body, transcriptional factors, cytoskeletal proteins, and RNA-binding proteins (Fig. 4). PIAS1 was previously shown to colocalize to PML nuclear body and to regulate oncogenic signaling through SUMOylation of PML and its gene translocation product PML-RARα associated with APL[26].

Interestingly, we also found that several putative PIAS1 substrates were associated with cytoskeletal organization including β-actin (ACTB), α-tubulin (TUBA1B), and VIM, in addition to several other IF proteins. The actin filaments, IFs, and microtubules that form the cytoskeleton of eukaryotic cells are responsible for cell division and motility. They also help establish cell polarity, which is required for cellular homeostasis and survival[30]. Moreover, one SUMO site on β-actin (ACTB) and two of the five SUMO sites on α-tubulin (TUBA1B) that were found at non-consensus motif regions were regulated by PIAS1, suggesting an important role for PIAS1 in substrate protein dynamics during SUMOylation. In addition, we evaluated the degree of evolutionary conservation of these modified lysine residues. Surprisingly, all SUMOylated lysine residues analyzed are highly conserved across different species (Supplementary Figs. 5 and 6). In our data, we found IFs to be the major targets of

PIAS1 among cytoskeletal proteins. Our results highlight that PIAS1 mediates the SUMOylation of the type III IF VIM protein at Lys-439 and Lys-445, both of which are located on the tail domain. Moreover, PIAS1 promotes the SUMOylation of several type V IF proteins (e.g., Prelamin A/C, Lamins B1, and B2) on their Rod domains (Supplementary Fig. 7).

**Validation of PIAS1 substrates by in vitro SUMOylation assays.** Next, we selected E3 SUMO-protein ligase NSE2 (NSMCE2) and prefoldin subunit 2 (PFDN2), which were identified in SUMO proteomic experiments as putative PIAS1 substrates for further validation. We performed in vitro SUMOylation assays to confirm that these sites were regulated by PIAS1. For the reconstituted in vitro SUMO assay, we incubated individual SUMO substrates with SUMO-activating E1 enzyme (SAE1/SAE2), UBC9, and SUMO3 with or without PIAS1 in the presence of ATP. We also used PCNA, a known PIAS1 substrate, as a positive control. After 4 h incubation at 37 °C, the western blottings of each substrate showed either single or multiple bands of higher molecular weight confirming the SUMOylated products. Separate LC-MS/MS experiments performed on the tryptic digests of the in vitro reactions confirmed the SUMOylation of NSMCE2 at residues Lys-90, Lys-107, and Lys-125, and PFDN2

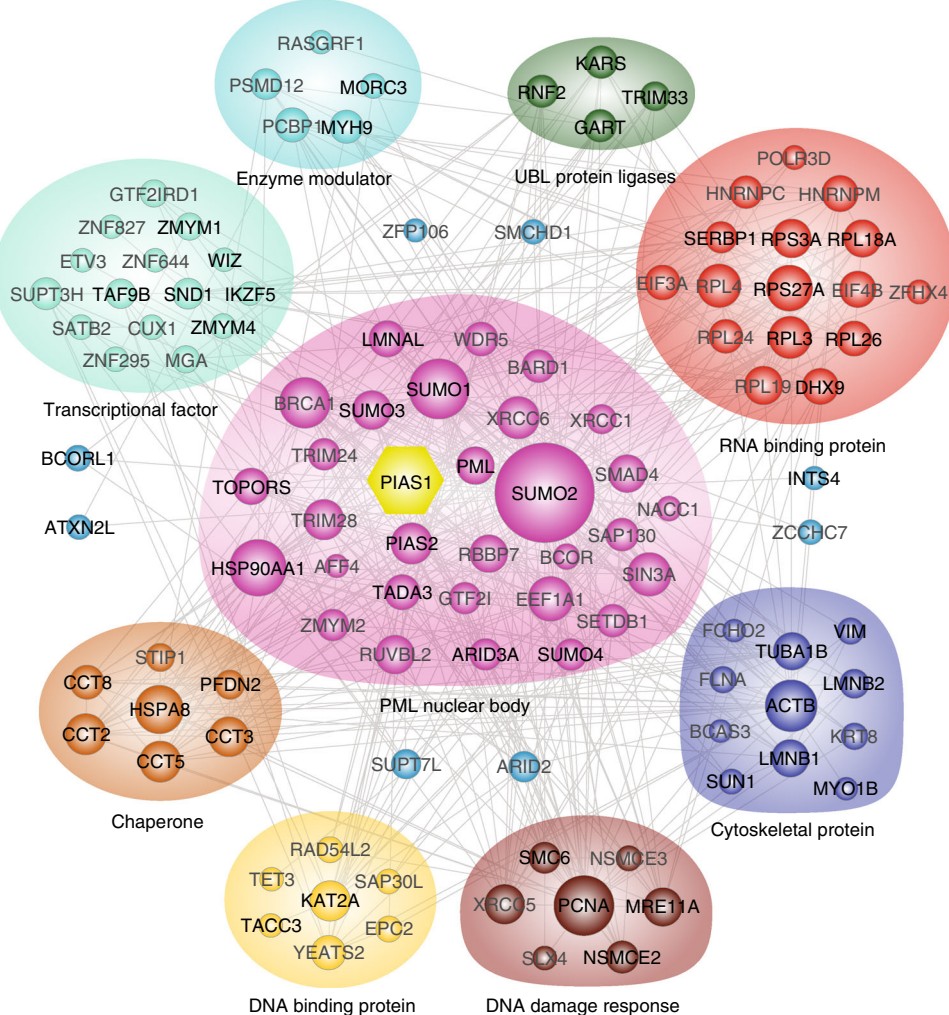

**Fig. 4 Protein–protein interaction network of PIAS1 substrates.** STRING network of PIAS1 substrates and their interacting partners. Proteins are grouped according to their GO terms.

at residues Lys-94, Lys-111, Lys-132, and Lys-136. Although UBC9 alone can SUMOylate these substrates, we noted an increasing abundance of SUMOylated proteins when PIAS1 was present, confirming that the E3 SUMO ligase enhanced the efficiency of the conjugation reaction (Supplementary Fig. 8a). Interestingly, several SUMOylation sites that were regulated by PIAS1 on both NSMCE2 and PFDN2 were not located within SUMO consensus motifs, further supporting the motif analysis of the large-scale proteomic data (Supplementary Fig. 4a).

Furthermore, we examined whether PIAS1 contributes to substrate SUMOylation in vitro using a cell-based assay. HEK293-SUMO3m cells were co-transfected with Flag-NSMCE2 or PFDN2 and Myc-PIAS1. Co-transfected cells were subjected to immunoprecipitation with anti-Flag agarose gel, followed by western blotting with an anti-His antibody. The SUMOylation of substrates was minimally detected when only transfecting Flag substrates. In contrast, overexpression of PIAS1 under the same experimental conditions led to a marked increase in the SUMOylation of these substrates (Supplementary Fig. 8b). These results further confirm our quantitative SUMO proteomics data and validate the proteins NSMCE2 and PFDN2 as bona fide PIAS1 substrates.

**PIAS1 SUMOylation promotes its recruitment to PML nuclear body.** Interestingly, our large-scale SUMO proteomic analysis

identified five SUMOylation sites on PIAS1 (Fig. 5a). Two of these sites (Lys-46 and Lys-56) are located in the SAP domain, and may regulate the interaction of PIAS1 with DNA[15]. We also identified two SUMOylated residues (Lys-137 and Lys-238) located within the PINIT domain of PIAS1, potentially affecting its subcellular localization[14]. The last SUMOylated site (Lys-315) of PIAS1 is located next to an SP-RING domain, which may alter the ligation activity of PIAS1[31]. Of note, PIAS1 contains a SIM and previous reports indicated that this ligase can localize to PML nuclear bodies in a SIM-dependent manner with SUMOylated PML[32]. As PML also contains a SIM motif, we were interested in three out of the five SUMO sites on PIAS1: Lys-137, Lys-238, and Lys-315. As PIAS1 is SUMOylated at several sites and PML contains a SIM, we surmised that reciprocal interactions could be mediated through SUMO-SIM binding. Accordingly, we constructed a PIAS1-GFP vector and used site-directed mutagenesis to specifically mutate the PIAS1 lysine residues that are SUMO-modified and are located within regions of PIAS1 that could interact with PML. As the SAP domain of PIAS1 is exclusively reserved for DNA binding, we excluded the SUMO-modified lysine residues in this domain when creating the mutant construct as it may affect its localization in a PML-independent manner. We therefore created the variant constructs by mutating the codons for Lys-K137, Lys-238, and Lys-315 to arginine codons. Several mutant genes were created, including single mutants of each site and the triple mutant (PIAS1-GFP-3xKR). These mutant

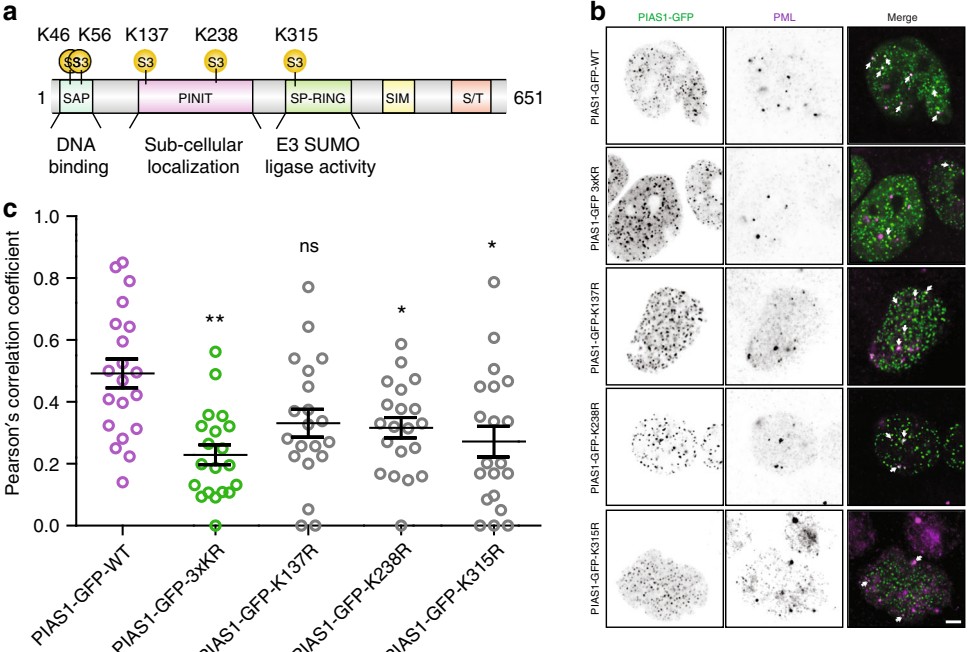

**Fig. 5 SUMOylation of PIAS1 promotes its PML localization. a** Distribution of SUMO sites identified on PIAS1. Three SUMO sites (K137, K237, and K315) were identified in the dataset. K137 and K238 are located in the PINIT domain, whereas K315 is located in the SP-RING domain. **b** HEK293-SUMOm cells were co-transfected with PIAS1-GFP-WT, PIAS1-GFP-3xKR, PIAS1-GFP-K137R, PIAS1-GFP-K238R, or PIAS1-GFP-K315R and immunofluorescence was performed with anti-PML. Scale bar: 2.5 μm. **c** Scatter graph showing the Pearson's correlation coefficient for the PIAS1–PML colocalization. Data represent the mean ± SD, error bars represent SD, ns nonsignificant, *p < 0.05, **p < 0.01, Student's t-test, n = 20 biologically independent cells/condition. Source data are provided as a Source Data file.

vectors were transfected into HEK293-SUMO3m cells and were used to study the effects of SUMOylation at the various lysine residues on the PIAS1-PML colocalization. As evidenced by the immunofluorescence studies, ~49% of PIAS1-GFP-WT colocalized with PML (Fig. 5b). Of all the single variants tested, significant changes in the colocalization of PIAS1 and PML were observed with the K238R and K315R alterations (Fig. 5c). However, we noted a >50% reduction in PIAS1-GFP–PML colocalization when all three sites were mutated, suggesting a possible cooperativity among these sites (Figs. 5b, c). This functional redundancy may be required to ensure the proper localization of PIAS1 to PML nuclear bodies under different biological context. Also, the cooperative nature of multiple SUMOylation events to enhance affinity has been noted before, where the affinity of RNF4 for SUMO dimers is tenfold higher than for the monomer[33].

The fact that the colocalization of PML and PIAS1-GFP was not totally abrogated for the triple mutant might be explained by residual interactions between PML and the PIAS1-GFP[32]. Indeed, we have shown that several sites on PIAS1 are SUMOylated, which aids to localize PIAS1 to the PML bodies. However, PML itself is also heavily SUMOylated on several lysine residues. Therefore, non-SUMOylated PIAS1 can still localize, albeit less efficiently, to PML nuclear bodies via the SIM that is located on PIAS1 and the SUMOylated moieties on PML. A similar phenomenon was reported by our group for the SUMO E2 protein UBC9[34]. Taken together, these experiments confirmed that colocalization of PIAS1 at PML nuclear bodies is partly mediated by the SUMOylation of PIAS1 at Lys-137, Lys-238, and Lys-315 residues.

**VIM SUMOylation promotes cell migration and motility.** VIM is predominantly found in various mesenchymal origins and

epithelial cell lines[35–37]. Increasing evidence shows that VIM plays key roles in cell proliferation[38], migration[39], and contractility[40]. Our data show that two SUMO sites on VIM are regulated by PIAS1, both of which are located on the tail domain and are highly conserved across different species (Fig. 6a). To confirm that VIM is selectively SUMOylated by PIAS1, we used CRISPR/Cas9 gene-editing technology with single guide RNA (sgRNA) specific to each of the four PIAS E3 ligases (e.g., PIAS1, PIAS2, PIAS3, and PIAS4) and a scrambled sgRNA for the transfection in HEK293-SUMO3m cells. KO and control HEK293-SUMO3 cells were cultured in triplicate using light, medium, and heavy SILAC media (Supplementary Fig. 9). Following NiNTA enrichment, SUMOylated proteins were digested on beads with trypsin and modified tryptic peptides were isolated by SUMO remnant immunoaffinity purification prior to targeted LC-MS/MS analyses using an inclusion list to detect and identify individual isotopically labeled SUMOylated peptides of VIM (e.g., ETNLDSLPLVDTHSK*R and TLLIK*TVETR, where * indicates the SUMOylation site). MS/MS spectra of isotopically labeled tryptic peptides were used to confirm identification of SUMOylated VIM peptides (Supplementary Fig. 10). We also analyzed by LC-MS/MS in data-dependent acquisition the tryptic peptides from the flow through proteins, to normalize protein abundance across the six different samples. These quantitative proteomics experiments revealed that PIAS1 selectively targeted the SUMOylation of VIM at K439 and K445.

To further investigate the function of PIAS1-mediated SUMOylation of VIM, we expressed a Flag-tagged vimentin K439/445R double mutant (Flag-VIM^mt) that is virtually refractory to SUMOylation in HeLa cells, and compared the functional effects to cells expressing the wild-type Flag-tagged vimentin (Flag-VIM^wt). We transfected Flag-VIM^wt and Flag-VIM^mt into HeLa cells and used the empty Flag vector as a negative control. At 48 h post transfection, the cells were

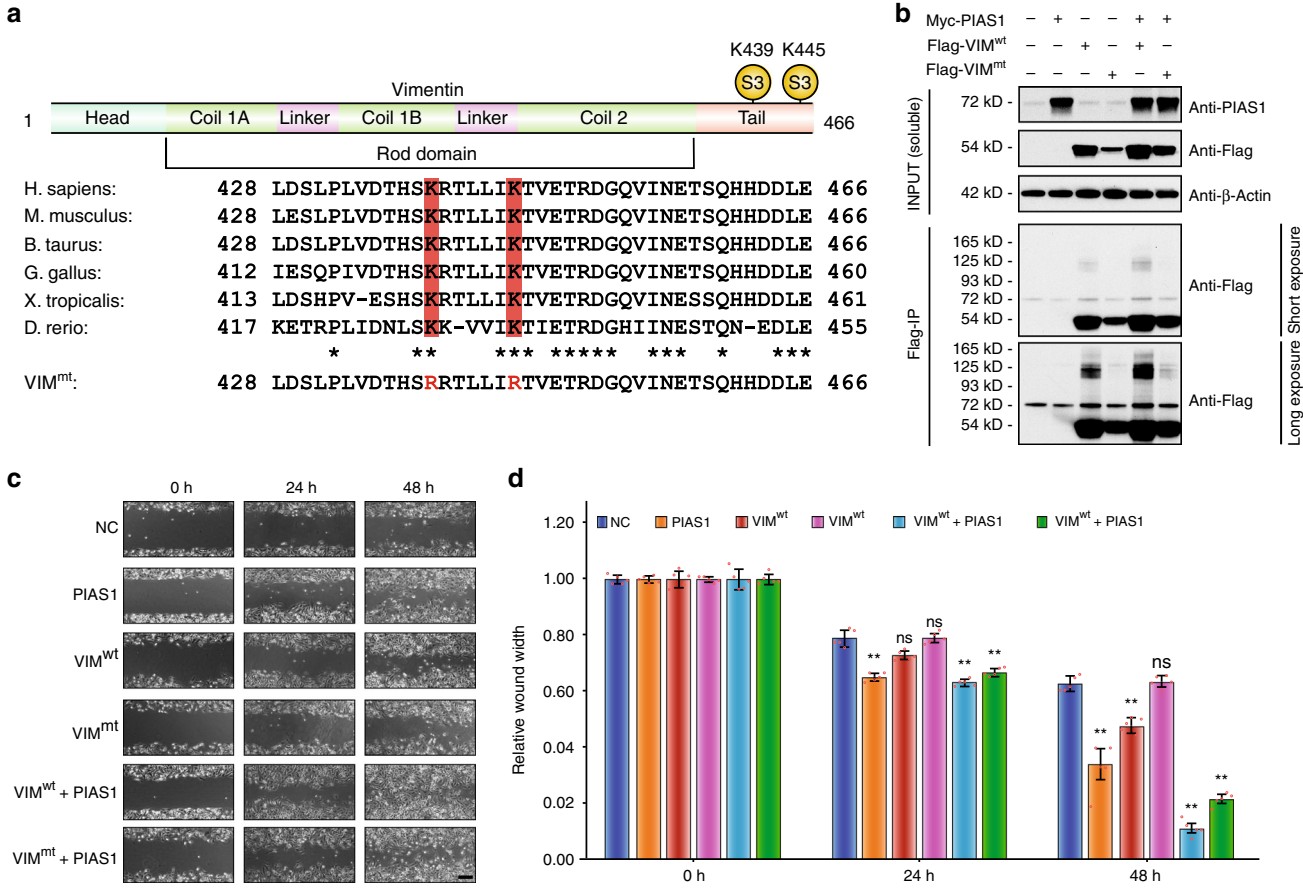

**Fig. 6 SUMOylation of vimentin is required for proper cell migration in HeLa cells. a** VIM sequence indicating SUMOylation sites at Lys-439 and Lys-445 localized in the tail domain and highly conserved across six different species. **b** Western blotting analysis of HeLa cells transfected with an empty Flag vector as a negative control, Wild-type Flag-tagged vimentin (Flag-VIM^wt), Flag-tagged vimentin K439/445R, double mutant (Flag-VIM^mt), with or without Myc-PIAS1. **c** Flag-VIM^mt expression inhibits proper cell migration and can be rescued by PIAS1 overexpression, as determined by a wound-healing assay. Scale bar: 200 μm. **d** Quantification of the wound-healing assay. Data represent the mean ± SD, error bars represent SD, ns nonsignificant, **$p <$ 0.01, Student's $t$-test, $n = 5$ biologically independent samples. Source data are provided as a Source Data file.

collected, lysed in 8 M urea, and protein pellets were separated on SDS-polyacrylamide gel electrophoresis (PAGE). The ensuing western blotting results show that protein abundance between VIM^wt and VIM^mt are similar; yet, the SUMO level on Flag-VIM^mt is undetectable (Fig. 6b). Moreover, transfecting PIAS1 considerably increased the level of SUMOylation of Flag-VIM^wt, supporting our proteomics experiments. Of note, transfecting PIAS1 along with Flag-VIM^mt promoted the SUMOylation of Flag-VIM^mt, albeit to a much lower degree than Flag-VIM^wt, improving the solubility Flag-VIM^mt.

We examined the effect of Flag-VIM^wt and Flag-VIM^mt expression on cell migration using the wound-healing assay. Flag-VIM^wt significantly promotes cell migration (Fig. 6c), which is in line with the results obtained in HepG2 cells[41]. However, Flag-VIM^mt alone conferred no effect on cell migration, while transfecting PIAS1 with Flag-VIM^mt rescued the phenotypic effect (Fig. 6c, d). These results indicate that SUMOylation of VIM plays a role in cell growth and migration, presumably by regulating VIF function and/or formation.

Next, we investigated the function of VIM SUMOylation on the dynamics assembly of IFs. Both Keratin and lamin A IFs formation and solubility have been reported to be regulated by SUMOylation, while such properties have yet to be uncovered for VIFs[42]. For example, the SUMOylation of lamin A at Lys-201, which is found in the highly conserved rod domain of the protein,

results in its proper nuclear localization[43]. Unlike lamin A SUMOylation, keratin SUMOylation is not detected under basal conditions. However, stress-induced keratin SUMOylation has been observed in mouse and human in chronic liver injuries. In addition, keratin monoSUMOylation is believed to increase its solubility, whereas hyperSUMOylation promotes its precipitation[44].

We surmised that SUMOylation on VIM would alter its solubility, akin to the properties observed for keratin. Accordingly, we transfected an empty Flag vector as a negative control, Flag-VIM^wt, and Flag-VIM^mt into HeLa cells and lysed the cells with a radioimmunoprecipitation assay (RIPA) buffer. Samples were fractionated into RIPA-soluble and -insoluble fractions. Western blot analysis of these samples shows that Flag-VIM^wt is preferentially located in the RIPA-soluble fraction, whereas Flag-VIM^mt resides more in the insoluble fraction (Fig. 7a). Clearly, SUMOylation of VIM drastically increases its solubility.

VIM filaments are also phosphorylated at multiple sites by several kinases[45] and this modification is associated with their dynamic assembly and disassembly[46–48]. In particular, phosphorylation sites located at the N-terminal head domain can impede the interaction between VIM dimer, thus preventing the formation of VIM tetramer necessary for the further assembly into filaments. Activation of Akt in soft-tissue sarcoma cells promotes the interaction of the head region of VIM and the tail

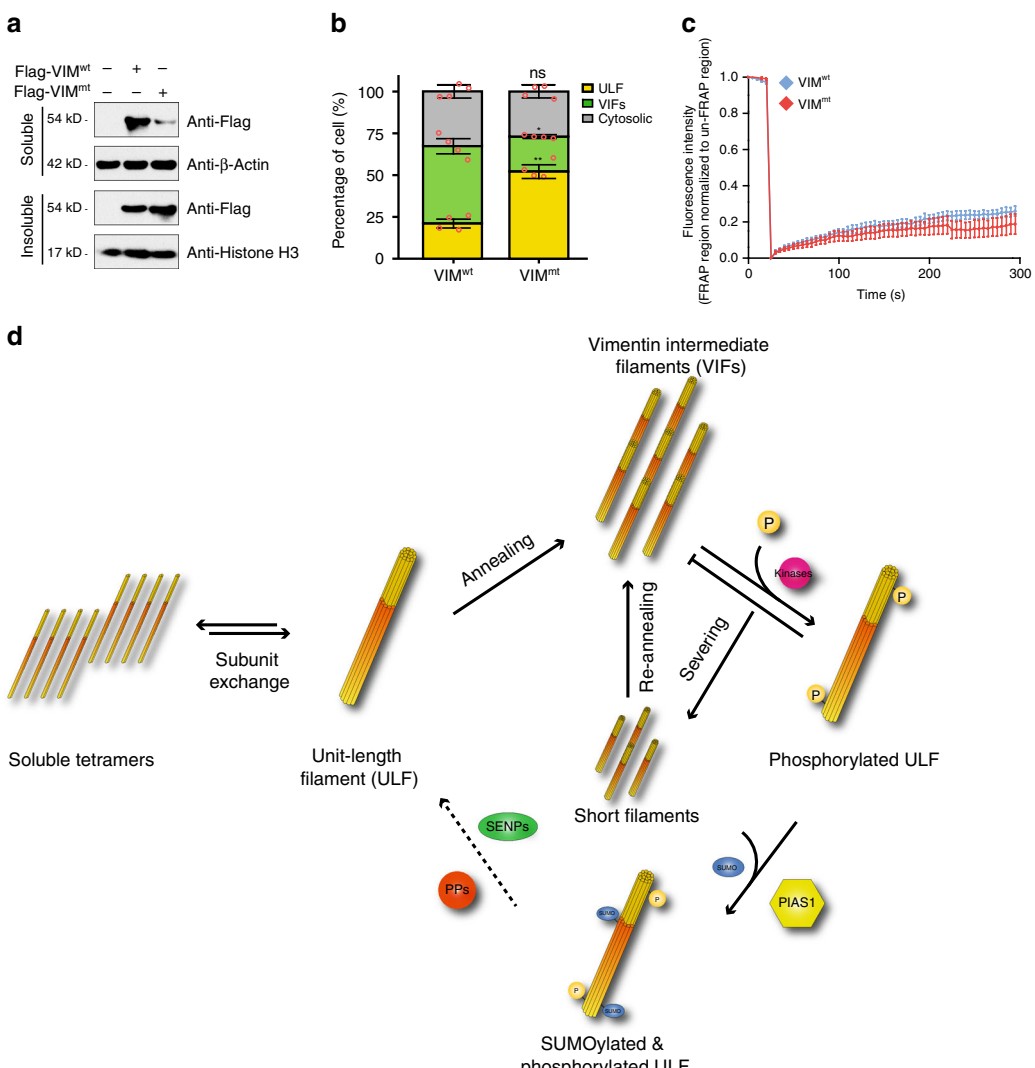

**Fig. 7 SUMOylation of vimentin regulates its dynamic assembly. a** HeLa cells were transfected with an empty Flag vector as a negative control, Flag-VIM[wt], and Flag-VIM[mt], and were separated into RIPA-soluble and -insoluble fractions. VIM protein levels were examined by western blotting. **b** MCF-7 cells were transfected with Emerald wild-type vimentin (VIM[wt]) and Emerald vimentin K439/445R, double mutant (VIM[mt]), and the proportion of unit-length filament (ULF), vimentin intermediate filament (VIF), and cytosolic vimentin between VIM[wt] and VIM[mt] were calculated under a microscope. Data represent the mean ± SD, error bars represent SD, ns nonsignificant, *$p < 0.05$, **$p < 0.01$, Student's $t$-test, $n = 4$ biologically independent samples. **c** Fluorescence recovery after photobleaching (FRAP) assay of Emerald wild-type vimentin (VIM[wt]) and Emerald vimentin K439/445R, double mutant (VIM[mt]) in MCF-7 cells. Line plot shows average fluorescence at each time point ± SD. Differences between values for VIM[wt] and VIM[mt] were not statistically significant at all time points by Student's $t$-test. $n = 7$ biologically independent cells. **d** Model of the VIM dynamic assembly and disassembly. Vimentin is maintained in equilibrium between unit-length filament (ULF) and soluble tetramers (subunit exchange step). Vimentin filaments elongate by end-to-end annealing of ULF to form mature vimentin intermediate filament (VIF; annealing step). The phosphorylation-dependent shortening of VIF (severing step) involves phosphorylation on Ser-39 and Ser-56 of vimentin by several kinases, including Akt1. The short filaments can re-anneal with another ULF to form new VIF (re-annealing step). However, the phosphorylated ULF is not amenable to the re-annealing process. These phosphorylated ULF products are subject to PIAS1-mediated SUMOylation, stimulating the dephosphorylation of the phosphorylated ULF, and subsequently re-enter to either subunit exchange process or VIF maturation process. Source data are provided as a Source Data file.

region of Akt, resulting in the phosphorylation of VIM at Ser-39, further enhancing cell motility and invasion[43].

Next, we sought to examine whether VIM SUMOylation alters its phosphorylation status, which could lead to changes in VIF dynamic assembly/disassembly. Accordingly, we separated protein extracts from Flag-VIM[wt] and Flag-VIM[mt] by SDS-PAGE, excised bands that corresponded to the soluble and insoluble VIM, and performed in-gel trypsin digestion followed by LC-MS/MS (Supplementary Fig. 11). We identified several phosphorylated serine residues and one phosphorylated threonine residue on VIM (Supplementary Data 3), which have also been reported in the literature (S5, S7, T20, S22, S26, S29, S39, S42, S51, S56, S66, S72, S73, S83, S226, T258, and S459). All sites, except those located on the C-terminal, were found to be hyperphosphorylated in the insoluble VIM[mt] compared with the wild-type counterpart. Interestingly, we observed an increase phosphorylation of S39 in the insoluble pellet of VIM[mt], a site known to be phosphorylated by Akt[43]. The observation that VIM is hyperphosphorylated at its N-terminus in Flag-VIM[mt] suggests a possible cross-talk between SUMOylation and phosphorylation.

To further understand how SUMOylation affects VIM dynamics in vivo, we transfected Emerald-VIM[wt] or Emerald-VIM[mt] vectors in VIM-null MCF-7 cells. VIM-null cells were employed to eliminate the contribution of endogenous VIM on

the VIM dynamics, which could mask the phenotypic effects of our mutant. Using fluorescence microscopy, we quantified the proportion of the various VIM structures for the two different Emerald-tagged constructs. We found three major forms of VIM in the cells, which in accordance with the literature, were categorized as cytosolic, unit-length filament (ULF) and VIFs (Supplementary Fig. 12). Statistical analysis of the proportion of the VIM structures revealed that the SUMO conjugation-deficient VIM (VIM$^{mt}$) promoted the formation of ULF with a concomitant reduction in VIF formation compared with its wild-type counterpart (VIM$^{wt}$) (Fig. 7b). Taken together, these results indicate that SUMOylation of VIM promotes the formation of VIFs from the ULF building blocks.

Although VIM filament growth primarily relies on elongation by the longitudinal annealing of ULFs via end-to-end fusion, recent studies suggest that the subunit exchange of tetramers within these filaments does occur[49]. To determine whether the SUMOylation of VIM affects the subunit exchange rate of these filaments, we performed fluorescence recovery after photobleaching (FRAP) assays (Supplementary Fig. 13). We monitored the recovery time of filament fluorescence up to 300 s after bleaching for both Emerald-VIM$^{wt}$ and Emerald-VIM$^{mt}$, and noted no statistical difference in recovery between constructs (Fig. 7c). This observation suggests that the SUMOylation of VIM is not involved in the subunit exchange of tetramers within filaments.

The model depicted in Fig. 7d combines the results from the proteomic, immunofluorescence, and FRAP assays, and describes the molecular mechanism of PIAS1-mediated control of VIM dynamics. Under physiological conditions, VIM is maintained in equilibrium between ULF and soluble tetramers. The formation of VIM ULF has been shown to occur spontaneously on the order of seconds in vitro showing that this arrangement is thermo-dynamically favorable and proceeds rapidly without the need for protein modifications[50]. VIM filaments elongate by end-to-end annealing of ULFs and eventually form mature VIFs. This is followed by the breakdown of VIFs by severing, which involves phosphorylation on several residues found on the N-term of VIM[45,51]. We show in this work that this hyper-phosphorylation occurs exclusively on the N-terminal of VIM in a SUMO-dependent mechanism. The truncated filaments can re-anneal with another ULF to form larger VIFs. However, the phosphory-lated ULF must be SUMOylated by PIAS1 to increase either the solubility or interaction with protein phosphatases, such as type-1 (PP1) and type-2A (PP2A) protein phosphatases, as shown by the large increase in VIM phosphorylation levels with the SUMO-deficient VIM construct[45]. These results suggest that the PIAS1-mediated SUMOylation of VIM stimulates the dephosphorylation of ULF and facilitate the re-entry of the ULF into the VIF maturation process by annealing on growing VIFs. This dynamic assembly and disassembly of VIFs thus involve the SUMOylation of VIM, a modification that also regulates the cell migration and motility (Fig. 6).

## Discussion

We report the functional effect of E3 SUMO ligase PIAS1 in HeLa cells and determined that PIAS1 not only promotes cell pro-liferation but also stimulates cell migration. PIAS1 has been extensively studied in other cancer lines, such as human prostate cancer, where PIAS1 expression is increased and enhanced pro-liferation through inhibition of p21[21]. In addition, other studies have also reported that PIAS1 may function as a tumor suppressor to regulate gastric cancer cell metastasis by targeting the mitogen-activated protein kinase signaling pathway[52]. Interleukin 11 was previously shown to reduce the invasiveness of HTR-8/SVneo cells via reduced ERK1/2 activation, PIAS1/3-mediated activated

STAT3 (Tyr-705) sequestration, and a decrease in PIAS1 expres-sion, leading to reduced expression of Fos and several major metalloproteinases (MMP2, MMP3, MMP9 and MMP23B)[53]. However, these studies were limited to individual PIAS1 targets to understand the regulation mechanism. These targeted approaches sufficed to answer specific questions about PIAS1-mediated SUMOylation, but lack the depth to fully elucidate the function of PIAS1. A systematic approach to establish the global properties of PIAS1 as an E3 SUMO ligase and how these SUMOylation events alter substrate function are missing and needed. Indeed, such a method was never conceived due to the complex nature of quantitative SUMO proteomics. Global SUMO proteome analyses are challenging due to the low abundance of protein SUMOylation and the extremely large remnant that is retained on the modified lysine residues upon tryptic digestion. Moreover, proteomic workflows that are currently available to study SUMOylation require two levels of enrichment, which adversely affects the reproducibility of SUMO site quantification.

We used a straightforward method for the identification of PIAS1 substrates by expanding on our previously described SUMO proteomics strategy[27]. This method combines SILAC labeling for reproducible quantitative proteomic analyses, E3 SUMO ligase protein overexpression, followed by SUMO rem-nant immunoaffinity enrichment. This workflow allows for the selective profiling of substrates and regulated SUMOylation sites of any E3 SUMO ligase. All the PIAS1 substrates identified in this work were analyzed using forward and reverse SILAC labeling under basal condition, which further increases the confidence of the identified substrates. Notably, we observed that PIAS1 over-expression has a global effect on protein SUMOylation (Fig. 3b). This is in part due to some PIAS1 substrates being directly involved in protein SUMOylation, such as PML, PIAS2, NSMCE2, and TOPORS. In addition, many of the identified substrates were found to participate in protein ubiquitination regulation, such as TRIM33 and RNF2, which may also affect the global protein SUMOylation through the interplay between SUMOylation and ubiquitination[27,54]. As for other closely related PIAS family members, the regulatory function of PIAS1 extend beyond the SP-RING-type SUMO ligase and can mediate protein interactions through non-covalent SUMO binding. Indeed, the SUMOylation of direct substrates of PIAS1 can promote the modifications of other targets of the same protein complexes. Also, the complex formation maybe facilitated via the SAP and PINIT domains of PIAS1 through the interactions with the chromatin or the subcellular localization of binding partners[9,14]. Thus, several factors affect protein SUMOlation, and some of the SUMOylation sites that increased upon PIAS1 overexpression may not be direct substrates of PIAS1, or may be SUMOylated to a different extent when PIAS1 is expressed at endogenous levels.

We identified five SUMOylation sites on PIAS1 itself (Lys-46, Lys-56, Lys-137, Lys-238, and Lys-315), suggesting a possible feedback mechanism that could keep SUMOylation levels in check. Our immunofluorescence studies show that SUMOylation of PIAS1 promotes its localization to PML nuclear bodies (Fig. 5b, c). Interestingly, a recent paper that studied the sub-strates of RNF4 identified PIAS1 as a substrate of this SUMO-targeted ubiquitin ligase[55]. Moreover, RNF4 is localized to PML nuclear bodies, where it ubiquitylates SUMOylated proteins for their subsequent proteasomal degradation. Elevated levels of cellular SUMOylation may lead to an increased SUMOylation of PIAS1, prompting its localization to PML nuclear bodies and its degradation by RNF4 in a ubiquitin-dependent manner. This feedback mechanism used to regulate global SUMOylation may not be reserved solely for PIAS1. Indeed, other members of the PIAS family, as well as NSE2 and TOPORS, have been found to

be SUMOylated at several lysine residues, while also being substrates of RNF4[55].

Notably, cytoskeletal proteins occupy a significant proportion of the identified PIAS1 substrates. Constituents of actin filaments, IFs, and microtubules were all found to be PIAS1 substrates. Interestingly, unlike UBC9 substrates that are typically SUMOylated on consensus motifs[6], the acceptor lysine residues found on these cytoskeletal proteins are highly conserved but are located in non-consensus sequence motif. These observations suggest that PIAS1 may act as an adaptor protein to change cytoskeletal protein turnover or dynamics by facilitating their SUMOylation. We uncovered that PIAS1 specifically SUMOylates K439 and K445 residues of VIM. This modification increased the solubility of VIM and is correlated with the uptake of ULF onto VIF in a phospho-dependent mechanism. VIM SUMOylation in turn favors cell proliferation and motility, which could lead to an increase in cancer cell aggressiveness. Although these findings could reveal the molecular mechanism of PIAS1-mediated VIM SUMOylation and its involvement in cancer cell aggressiveness, additional evidence is required to further understand the function of PIAS1-mediated SUMOylation on the other cytoskeletal proteins and how these cytoskeletal proteins collaborate during cell migration.

## Methods

**Cell culture, vector construction, and gene knockout.** Human cervical cancer cell line (HeLa) (ATCC® CCL-2™, Cedarlane) and HEK293 stably expressing the 6xHis-SUMO3-Q87R/Q88N mutant (HEK293-SUMO3m)[27] were cultured in Dulbecco's modified Eagle's medium (DMEM) (HyClone) supplemented with 10% fetal bovine serum (FBS; Wisent), 1% L-glutamine (Thermo Fisher Scientific), and 1% penicillin/streptomycin (Invitrogen) in 5% $CO_2$ at 37 °C.

The mammalian expression vector for Myc-PIAS1 was constructed by inserting the full-length cDNAs into pcDNA3.0-Myc. The mammalian expression vector for PIAS1-GFP-WT was constructed by cloning full-length PIAS1 cDNA into pcDNA3.1-c-GFP10. The mammalian expression vectors pReceiver-M11 (Flag-NSMCE2, Flag-PFDN2, Flag-VIM, and Flag-empty control) were purchased from Genecopoeia, Inc. (Rockville, MD). The mammalian expression vector Emerald-VIM was purchased from addgene (#54300). The PIAS1-GFP-K137R, PIAS1-GFP-K238R, PIAS1-GFP-K315R and PIAS1-GFP-3XKR, Flag-VIM^mt, and Emerald-VIM^mt plasmids were generated by site-directed mutagenesis using the GeneArt Site-Directed Mutagenesis System, according to the instructions of the manufacturer (Invitrogen™). The PIAS CRISPR/Cas9-based gene KO vectors pCRISPR were also purchased from Genecopoeia, Inc. (Rockville, MD). Primer sequences used for vector construction, site-directed mutagenesis, and sgRNA sequences used for CRISPR/Cas9 gene KO are listed below: PIAS1_F: 5′-GGGTAC CATGGCGGACAGTGCGGAAC-3′, PIAS1_R: 5′-GGAATTCTCAGTCCAATG AAATAATGTCTGG-3′, PIAS1_K137R_F: 5′-GTCCATCCGGATATAAGACT TCAAAAATTACCA-3′, PIAS1_K137R_R: 5′-TGGTAATTTTTGAAGTCTTA TATCCGGATGGAC-3′, PIAS1_K238R_F: 5′-TACCTTCCACCTACAAGAAAT GGCGTGGAACCA-3′, PIAS1_K238R_R: 5′-TGGTTCCACGCCATTTCTTGT AGGTGGAAGTA-3′, PIAS1_K315R_F: 5′-GCTTTAATTAAAGAGAGGTTGA CTGCGGATCCA-3′, PIAS1_K315R_R: 5′-CGGATCCGCAGTCAACCTCTCT TTAATTAAAGC-3′, VIM_K439R_F: 5′-GTTGATACCCACTCAAGAAGGA CACTTCTGATT-3′, VIM_K439R_R: 5′-AATCAGAAGTGTCCTTCTTGAG TGGGTATCAAC-3′, VIM_K445R_F: 5′-AGGACACTTCTGATTAGGACG GTTGAAACTAGA-3′, VIM_K445R_R: 5′-TCTAGTTTCAACCGTCCTAAT CAGAAGTGTCCT-3′, PIAS1-sgRNA: 5′-TTCTGAACTCCAAGTACTGT-3′, PIAS2-sgRNA: 5′-CAAGTATTACTAGGCTTTGC-3′, PIAS3-sgRNA: 5′-GCC CTTCTATGAAGTCTATG-3′, PIAS4-sgRNA: 5′-GGCTTCGCGCCGTAGT CTTAG-3′, and scrambled sgRNA: 5′-GGCTTCGCGCCGTAGTCTTA-3′.

For transient transfection, cells were transfected with 1 µg plasmid per million cells using JetPrime Reagent (Polyplus transfection) according to the manufacturer's protocol. Cells were collected 36 h or 48 h after transfection for further experiments and protein overexpression was confirmed by western blotting.

For PIAS gene KO, cells were transfected with 1 µg plasmid per million cells using JetPrime Reagent (Polyplus transfection) according to the manufacturer's protocol. Cells were sorted by fluorescence-activated cell sorting based on Red fluorescent proteins-mCherry signals 48 h after transfection. mCherry-positive cells were cultured and expended for another week (Supplementary Fig. 1). PIAS KO efficiency was confirmed by western blotting.

**Cell proliferation assay.** The cell proliferation assay was carried out using WST-1 Cell Proliferation Assay Kit (Roche). Forty-eight hours post-transfected cells or KO cells were seeded into 96-well plates at a density of 1000 cells/well. WST-1 reagents were added to each well at time point 0 h, 24 h, and 48 h, and cells were further

incubated at 37 °C for 1 h. The absorbance was measured on a microplate Reader Infinite® M1000 PRO (TECAN) with a test wavelength at 450 nm and a reference wavelength at 630 nm The relative numbers of viable cells were estimated by subtracting the 630 nm background absorbance from 450 nm measurements.

**Cell migration assay.** Cell motility was evaluated by using a wound-healing assay as follows: 36 h post transfection, cells were collected by a brief trypsinization and were seeded in Ibidi wound-healing two-well Culture Inserts (Ibidi) into 24-well plates. Cells were grown to confluence in DMEM containing 10% FBS for another 12 h before the Ibidi wound-healing two-well Culture Inserts were removed. The cells were washed twice with phoshate-buffered saline (PBS) to remove the cell debris and grown in DMEM containing 1% FBS. The cell migration into the gap area was observed and photographed at time point 0 h, 24 h, and 48 h. Closure of the gap was measured using a phase-contrast microscope. Wound-healing was analyzed using "MRI Wound Healing Tool" plugin in ImageJ and estimated as a percentage of the starting wound area.

**SILAC labeling and protein extraction.** HEK293-SUMO3m cells were grown in DMEM (Thermo Fisher Scientific) containing light (0Lys, 0Arg), medium (4Lys, 6Arg), or heavy (8Lys, 10Arg) isotopic forms of lysine and arginine (Silantes) for at least six passages, to ensure full labeling. For each triple SILAC experiment, the control channel was transfected with an empty-pcDNA3.0-Myc vector, while the other two channels were transfected with the Myc-PIAS1 plasmid. Similar conditions were used for the culture of PIAS KO cells using the combination scheme described in Supplementary Fig. 9. After 48 h transfection, an equal amount of cells from each SILAC channel were combined and washed twice with ice-cold PBS, lysed in NiNTA denaturing incubation buffer (6 M Guanidinium HCl, 100 mM $NaH_2PO_4$, 20 mM 2-Chloroacetamide, 5 mM 2-Mercaptoethanol, 10 mM Tris-HCl pH 8) and sonicated. Protein concentration was determined using micro Bradford assay (Bio-Rad).

**Protein purification, digestion, and desalting.** For each replicate, 16 mg of TCE were incubated with 320 µL of NiNTA beads (Qiagen) at 4 °C. After 16 h incubation, NiNTA beads were washed once with 10 mL of NiNTA denaturing incubation buffer, 5 times with 10 mL of NiNTA denaturing washing buffer (8 M urea, 100 mM $NaH_2PO_4$, 20 mM imidazole, 5 mM 2-Mercaptoethanol, 20 mM Chloroacetamide, 10 mM Tris-HCl pH 6.3), and twice with 10 mL of 100 mM ammonium bicarbonate. Protein concentration was determined using micro Bradford assay (Bio-Rad). Protein digestion on beads was carried out using a ratio 1:50 sequencing grade-modified trypsin (Promega): protein extract in 100 mM ammonium bicarbonate at 37 °C overnight. Proteins from the flowthrough of the NiNTA purification were trypsinized using a ratio 1:50 sequencing grade-modified trypsin (Promega) at 37 °C overnight for further proteome analyses. To quench the reaction, 0.1% trifluoroacetic acid (TFA) was added. The solution was desalted on hydrophilic–lipophilic balance cartridges (1 cc, 30 mg) (Waters) and eluted in LoBind tubes (Eppendorf) before being dried down by Speed Vac.

**SUMO peptide enrichment.** PureProteome protein A/G magnetic beads (Millipore) were equilibrated with anti-K (NQTGG) antibody (UMO-1-7-7, Abcam) at a ratio of 1:2 (v/w) for 1 h at 4 °C in PBS. Saturated beads were washed three times with 200 mM triethanolamine pH 8.3. For crosslinking, 10 µl of 5 mM dimethyl pimelimidate in 200 mM triethanolamine pH 8.3 was added per microliter of slurry and incubated for 1 h at room temperature (RT). The reaction was quenched for 30 min by adding 1 M Tris-HCl pH 8 to a final concentration of 5 mM. Crosslinked beads were washed three times with ice-cold PBS and once with PBS containing 50% glycerol. The tryptic digests were resuspended in 500 µl PBS containing 50% glycerol and supplemented with crosslinked anti-K-(NQTGG) at a ratio of 1:2 (w/w). After 1 h incubation at 4 °C, anti-K-(NQTGG) antibody-bound beads were washed three times with 1 ml of 1 × PBS, twice with 1 ml of 0.1 × PBS, and once with double-distilled water. SUMO peptides were eluted three times with 200 µl of 0.2% formic acid in water and dried down by Speed Vac.

**SCX fractionation.** Enriched SUMO peptides or tryptic peptides were reconstituted in water containing 15% acetonitrile (ACN) and 0.2% formic acid and loaded on conditioned SCX StageTips (Thermo Fisher Scientific). Peptides were eluted with ammonium formate pulses at 50, 100, 300, 600, and 1500 mM in 15% ACN, pH 3. Eluted fractions were dried down by Speed Vac and stored at −80 °C for MS analysis.

**Sample fractionation and in-gel digestion.** Cell pellets were resuspended in 5 pellet volumes of ice-cold RIPA buffer. The lysate was spun at $13,000 \times g$ for 10 min to separate the extract into RIPA-soluble and -insoluble fractions. The soluble and insoluble VIM samples were separated on a 4–12% SDS-PAGE (Bio-rad) and the proteins were visualized by Coomassie staining. The gel lane around VIM corresponding position (54 kDa) was cut and then diced into three ~1 mm cubes. During the process of in-gel digestion, the gel pieces were first destained completely using destaining solution (50% $H_2O$, 40% methanol, and 10% acetic acid). Then, the gel pieces were dehydrated by washing several times in 50% ACN until the gel

pieces shriveled and looked completely white. The proteins were reduced in 10 mM dithiothreitol at 56 °C for 30 min, alkylated in 55 mM chloroacetamide at RT in the dark for 30 min, and digested overnight with 300 ng of sequencing grade-modified trypsin in 50 mM ammonium bicarbonate. The supernatants were transferred into Eppendorf tubes and the gel pieces were sonicated twice in extraction buffer (67% ACN and 2.5% TFA). Finally, the peptide extraction and the initial digest solution supernatant were combined and then dried down using a Speed Vac and stored at −80 °C for MS analysis.

**Mass spectrometry analysis**. For PIAS1 substrates identification and proteome analysis, peptides were reconstituted in water containing 0.2% formic acid and analyzed by nanoflow-LC-MS/MS using an Orbitrap Fusion Mass spectrometer (Thermo Fisher Scientific) coupled to a Proxeon Easy-nLC 1000. Samples were injected on a 300 μm ID × 5 mm trap and separated on a 150 μm × 20 cm nano-LC column (Jupiter C18, 3 μm, 300 Å, Phenomenex). The separation was performed on a linear gradient from 7 to 30% ACN and 0.2% formic acid over 105 min at 600 nL/min. Full MS scans were acquired from m/z 350 to m/z 1500 at resolution 120,000 at m/z 200, with a target automatic gain control (AGC) of 1E6 and a maximum injection time of 200 ms. MS/MS scans were acquired in HCD mode with a normalized collision energy of 25 and resolution of 30,000 using a Top 3 s method, with a target AGC of 5E3 and a maximum injection time of 3000 ms. The MS/MS triggering threshold was set at 1E5 and the dynamic exclusion of previously acquired precursor was enabled for 20 s within a mass range of ±0.8 Da.

For targeted LC-MS/MS analyses of PIAS KO cells generated through CRISPR/Cas9 gene KO technology, SUMO peptides obtained from SUMO peptide enrichment were analyzed using an inclusion list to detect and identify each isotopolog of the VIM SUMOylated peptides at K439 (e.g., ETNLDSLPLVDTHSK*R) and K445 (e.g., TLLIK*TVETR). The SUMO peptides were reconstituted in water containing 0.2% formic acid and analyzed on an Orbitrap Q Exactive HF system (Thermo Fisher Scientific) coupled to a Proxeon Easy-nLC 1000. Samples were injected on a 300 μm ID × 5 mm trap and separated on a 150 μm × 20 cm nano-LC column (Jupiter C18, 3 μm, 300 Å, Phenomenex). The separation was performed on a linear gradient from 7 to 30% ACN and 0.2% formic acid over 105 min at 600 nL/min. The mass spectrometer was operated in a targeted-MS$^2$ acquisition mode with a maximum injection time of 1000 ms, 1 microscan, 30,000 resolution, 2E5 AGC target, 1.6 m/z isolation window, and 25% normalized collision energy.

For in-gel digested sample analyses, peptides were reconstituted in water containing 0.2% formic acid and analyzed on an Orbitrap Q Exactive HF system (Thermo Fisher Scientific) coupled to a Proxeon Easy-nLC 1000. Samples were injected on a 300 μm ID × 5 mm trap and separated on a 150 μm × 20 cm nano-LC column (Jupiter C18, 3 μm, 300 Å, Phenomenex). The separation was performed on a linear gradient from 7 to 30% ACN and 0.2% formic acid over 105 min at 600 nL/min. Full MS scans were acquired from m/z 350 to m/z 1200 at resolution 120,000 at m/z 200, with a target AGC of 5E6 and a maximum injection time of 50 ms. The precursor isolation window is set to 1.6 m/z with an offset of 0.3 m/z. MS/MS scans were acquired in HCD mode with a normalized collision energy of 25 and resolution of 30,000 using a TopN = 5 method, with a target AGC of 2E4 and a maximum injection time of 1000 ms.

**Data processing**. The SUMO proteome MS data were analyzed using MaxQuant (version 1.5.3.8)[56,57]. MS/MS spectra were searched against UniProt/SwissProt database (http://www.uniprot.org/) including Isoforms (released on 10 March 2015). The maximum missed cleavage sites for trypsin was set to 2. Carbamidomethylation (C) was set as a fixed modification and acetylation (Protein N-term), oxidation (M), deamination (NQ), and NQTGG (K) were set as variable modifications. The option match between runs was enabled to correlate identification and quantification results across different runs. The false discovery rate for peptide, protein, and site identification was set to 1%. SUMO sites with a localization probability of >0.75 were retained.

The flow through proteome MS data were searched with PEAKS X engine (Bioinformatics Solutions, Inc.) against the UniProt/SwissProt database (http://www.uniprot.org/) released on 5 June 2019. The precursor tolerance was set to 10 p.p.m. and fragment ion tolerance to 0.01 Da. The maximum missed cleavage sites for trypsin was set to 2. Carbamidomethylation (C) was set as a fixed modification, and oxidation (M), deamination (NQ) and NQTGG (K), $^2$H$_4$-lysine and $^{13}$C$_6$-arginine (SILAC medium), and $^{15}$N$_2$$^{13}$C$_6$-lysine and $^{15}$N$_4$$^{13}$C$_6$-arginine (SILAC heavy) were set as variable modifications with a maximum of five modifications per peptide. The false discovery rate for peptides was set to 1% with decoy removal. Proteins were quantified with ≥2 unique peptides. The relative change in protein abundance across samples was determined using the PEAKS X software. The targeted-MS data were analyzed using Skyline version 19.1, MaccCoss Lab Software, Seattle, WA (https://skyline.ms/wiki/home/software/Skyline/page.view?name=default), fragment ions for each targeted mass were extracted, and peak areas were integrated. Fold change ratios between the control and KO samples were calculated based on peak areas after normalizing peak intensities using the normalization factor determined from the proteome analysis.

The in-gel digested MS data were analyzed using MaxQuant (version 1.5.3.8)[56,57]. MS/MS spectra were searched against UniProt/SwissProt database (http://www.uniprot.org/) including Isoforms (released on 10 March 2015). The

maximum missed cleavage sites for trypsin was set to 2. Carbamidomethylation (C) was set as a fixed modification, and acetylation (Protein N term), oxidation (M), deamination (NQ) and phosphorylation (STY) were set as variable modifications. The false discovery rate for peptide, protein, and site identification was set to 1%. Phospho sites with a localization probability of >0.75 were retained.

**Bioinformatics analysis**. Classification of identified PIAS1 substrates was performed using PANTHER (Protein Analysis Through Evolutionary Relationships) (http://www.pantherdb.org), which classifies genes and proteins by their functions[58,59]. The identified PIAS1 substrates were grouped into the biological process, molecular function, and cellular component classes against the background of quantified SUMOylome using DAVID Bioinformatics Resources 6.7[60]. The aligned peptide sequences with ±6 amino acids sounding the modified lysine residue obtained in Andromeda were submitted to IceLogo[61]. For peptide sequences corresponding to multiple proteins, only the leading sequence was submitted. The secondary structures surrounding the PIAS1-regulated SUMO sites were investigated using NetSurfP-1.1[62]. INTERPRO Protein Domains Analysis and Protein–Protein Interaction (PPI) network of identified PIAS1 substrates were built by searching against the STRING database version 9.1[63,64]. All predictions were based on experimental evidence with the minimal confidence score of 0.4, which is considered as the highest confidence filter in STRING. PPI networks were then visualized by Cytoscape v3.5.1[65,66].

**Recombinant protein purification and in vitro SUMO assay**. The bacterial expression vectors pReceiver-B11 for His-NSMCE2 and His-PFDN2 recombinant protein expression were purchased from Genecopoeia, Inc. (Rockville, MD). pReceiver-B11 was individually transformed into ArcticExpress Competent Cells (#230191), which were derived from *Escherichia coli* B strains (Agilent Technologies). Transformed bacteria were cultured in Luria-Bertani medium until an optical density 600 of 0.5. Isopropyl β-D-1-thiogalactopyranoside (Bioshop) was added to the culture at a final concentration of 1 mM and the desired expression of the protein was inducted for 24 h at 13 °C. Collected bacterial pellets were lysed and sonicated in a solution containing 500 mM NaCl, 5 mM imidazole, 5 mM β-mercaptoethanol, 50 mM Tris-HCl pH 7.5. His-tagged proteins were purified on NiNTA beads (Qiagen). Purified proteins were eluted with 500 mM NaCl, 250 mM imidazole, 5 mM β- mercaptoethanol, 50 mM Tris-HCl pH 7.5, and concentrated on 3 kDa Ultra centrifugal filters (Amicon). Recombinant human PCNA protein was a generous gift from Dr Alain Verreault (University of Montreal, Canada). The in vitro SUMO assay was carried out in a buffer containing 5 mM MgCl$_2$, 5 mM ATP, 50 mM Tris-HCl pH 7.5. The enzymes responsible for the SUMOylation reaction were added at the following concentrations: 0.1 μM SAE1/2, 1 μM UBC9, 10 μM SUMO3, and 60 nM PIAS1. The reactions were incubated at 37 °C for 4 h and analyzed by western blotting.

**Cell-based in vitro SUMO assay and immunoprecipitation**. HEK293-SUMO3m cells or HeLa cells were co-transfected with Myc-PIAS1, Flag-NSMCE2, Flag-PFDN2, or Flag-VIM. At 48 h post transfection, cells were collected and lysed with 1 ml of Triton lysis buffer (150 mM NaCl, 0.2% Triton X-100, 1 mM EDTA, 10% Glycerol, 50 mM Tris-HCl pH 7.5) supplemented with protease inhibitor cocktail (Sigma Aldrich) and phosphatase inhibitor cocktail (Sigma Aldrich) at 4 °C for 15 min with gentle rocking. TCE was incubated with anti-Flag M2 Affinity Agarose Gel (Sigma Aldrich) with gentle rocking at 4 °C for overnight. Immunoprecipitates were then washed three times with cold Triton lysis buffer and were analyzed by western bloting.

**Western blotting**. TCE prepared in Triton lysis buffer were diluted in Laemmli buffer (10% (w/v) glycerol, 2% SDS, 10% (v/v) 2-mercaptoethanol, and 62.5 mM Tris-HCl pH 6.8), boiled for 10 min, and separated on a 4–12% SDS-PAGE (Bio-rad) followed by transfer onto nitrocellulose membranes. Before blocking the membrane for 1 h with 5% non-fat milk in Tris-buffered saline with Tween 20 (TBST), membranes were briefly stained with 0.1% Ponceau-S in 5% acetic acid to represent total protein content. Membranes were subsequently incubated overnight with a 1:1000 dilution of antibodies at 4 °C. Membranes were then incubated with peroxidase-conjugated Rabbit anti-mouse IgG (Light Chain Specific) (58802S, Cell Signaling Technology) or Goat-anti-rabbit IgG (7074S, Cell Signaling Technology) for 1 h at RT, at a 1:5000 dilution. Membranes were washed three times with TBST for 10 min each and revealed using ECL (GE Healthcare) as per the manufacturer's instructions. Chemiluminescence was captured on Blue Ray film (VWR). The following antibodies were used for western blot analyses: rabbit anti-Flag Antibody (F7425, Sigma Aldrich), mouse anti-Flag Antibody (F3165, Sigma Aldrich), rabbit anti-PIAS1 Antibody (3550S, Cell Signaling Technology), rabbit anti-PIAS2 Antibody (ab126601, Abcam), rabbit anti-PIAS3 Antibody (9042S, Cell Signaling Technology), rabbit anti-PIAS4 Antibody (4392S, Cell Signaling Technology), rabbit anti-Myc-Tag Antibody (2278S, Cell Signaling Technology), rabbit anti-His-Tag Antibody (SAB4301134, Sigma Aldrich), rabbit anti-α-Tubulin Antibody (2144S, Cell Signaling Technology), rabbit anti-β-Actin Antibody (4970S, Cell Signaling Technology), and rabbit anti-Histone H3 Antibody (4499S, Cell Signaling Technology).

**Fluorescence imaging and colocalization analysis**. HEK-SUMO3m cells were plated on 12 mm-diameter coverslips until they reached the desired density level and then transfected with the desired plasmids for 48 h. Cells were fixed in 4% paraformaldehyde in PBS for 15 min, followed by a 2 min permeabilization with 0.1% Triton X-100 in PBS and saturation with 2% bovine serum albumin (BSA) in PBS for 15 min. Cells were incubated with the rabbit anti-PML antibody (sc-5621, santa cruz) with a 1:100 dilution for 1 h at 37 °C, rinsed and incubated with secondary antibodies conjugated to Alexa Fluor 555 Phalloidin (8953 S, Cell Signaling Technology) with a 1:250 dilution and DAPI (4′,6-diamidino-2-phenylindole; D9542, Sigma Aldrich) with a 1:10,000 dilution for 1 h at RT. Both primary and secondary antibodies are diluted in PBS/BSA 2%. To reach sub-diffraction resolution, images from fixed samples were acquired with the Airyscan detector of a Zeiss LSM 880 confocal equipped with a ×63/1.43 Plan Apochromat oil-immersion objective. The number of PIAS1 and PML-positive structures were automatically detected and assessed by using the "particle analysis" tool in ImageJ. The Pearson's correlation coefficient was analyzed using ImageJ.

**Fluorescence recovery after photobleaching assay**. MCF-7 cells (ATCC® HTB-22™, Cedarlane) were plated in μ-Dish 35 mm dishes (Ibidi) until they reached the desired density level and then transfected with the Emerald-VIM$^{wt}$ or Emerald-VIM$^{mt}$ plasmids for 48 h. The FRAP assays were conducted on a LSM 880 confocal microscope equipped with a thermostatized chamber at 37 °C. The VIM Emerald-expressing cells were detected using a GaAsp detector. Bleaching was done by combining "Time", "Bleach," and "Region" modes on Zen software from Zeiss. Briefly, five pre-bleach images were taken every 5 s, after which five pulses of a 488 nm laser were applied to bleach an area of 25 × 2 μm. Post-bleach images were acquired every 5 s for 5 min. For fluorescence recovery analysis, the intensity in the bleached region was measured varying time points with "Frap profiler" plugin in ImageJ. Bleach data were normalized to unbleached regions for all the time points and expressed in arbitrary units in the recovery graphs.

**Quantification of VIM organization**. Fixed images of cells expressing Emerald-VIM$^{wt}$ or Emerald-VIM$^{mt}$ were taken using a LSM 880 confocal microscope. The "title" mode in the Zen software from Zeiss was used to cover a large area of cells (2.13 mm × 2.13 mm). To avoid localization and conformation artefacts due to expression levels, only the cells expressing Emerald-VIM at an intermediate level were evaluated using the threshold module in ImageJ. The same manual threshold was used for all conditions. Cells were tabulated using the "Cell counter" plugin in ImageJ into 4 categories: ULF, VIFs, cytosolic, and total cells. Category specifications were performed manually and subjectively. A representative image of each category is shown in Supplementary Fig. 12.

**Statistical analysis**. Statistical analysis was carried out to assess differences between experimental groups. Data were presented as the means ± SD. Statistical significance was analyzed by the Student's t-tests. $p < 0.05$ was considered to be statistically significant; $*p < 0.05$ and $**p < 0.01$.

**Reporting summary**. Further information on research design is available in the Nature Research Reporting Summary linked to this article.

## Data availability

The mass spectrometry proteomics data have been deposited to the ProteomeXchange Consortium (http://proteomecentral.proteomexchange.org) via the PRIDE partner repository with the dataset identifier PXD011932. The source data underlying Figs. 1a–f, 2b, 3d, e, 5c, 6b, 6d, and 7a–c, and Supplementary Figs. 4b, 8a, b, 9b, 9d, and 11 are provided as a Source Data file. All other data are available from the corresponding author upon request.

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

## Acknowledgements

This work was funded in part by the Natural Sciences and Engineering Research Council (NSERC) (P.T.; RGPIN-2018-04193) and the Canadian Institute for Health Research (CIHR) (G.E.; PJT 148943, PJT 148560). C.L. was supported by a scholarship from Fonds de recherche du Québec – Nature et technologies (FRQNT). The Institute for Research in Immunology and Cancer (IRIC) receives infrastructure support from Genome Canada, the Canadian Center of Excellence in Commercialization and Research, the Canadian Foundation for Innovation, and the Fonds de recherche du Québec - Santé (FRQS).

## Author contributions

C.L., F.P.M., C.P., C.M.P., T.N. and L.E.A.D. performed experiments and analyzed data. C.L., F.P.M. and P.T. wrote the manuscript. G.E. and P.T. developed the concept and managed the project.

## Competing interest

The authors declare no competing interests.
