## [Peer Review File · Nature Communications]

Reviewers' comments:

Reviewer #1 (Remarks to the Author):

Protein Inhibitor of Activated STAT 1 (PIAS1), with or without its SUMO ligase function, has been implicated in several important cellular pathways. There is also interesting evidence showing that the PIAS1 is overexpressed in several human cancers. Thibault's group has here used quantitative proteomics to identify PIAS1's targets in a proteome-wide fashion in HEK293 cells. This approach led to identification of novel, putative PIAS1's SUMO ligase substrates, including vimentin, involved in cell migration and motility. This is an interesting work utilizing state-of-the-art SUMO proteomics approaches. However, the actual role of PIAS1's SUMO ligase function in the promotion of cell proliferation and mobility remains unclear and requires clarification. To strengthen the message of this manuscript, a SUMO ligase function-mutated/dead version of PIAS1 (e.g. SP-RING 2xCys→Ser point mutant) should be tested in parallel with wtPIAS1 in HeLa cell-based proliferation, cell migration and "invasion" assays of experiments Figure 1 and Figure 6.

Reviewer #2 (Remarks to the Author):

The article provides a novel quantitative approach for SUMO proteome analysis and identified new SUMO sites on hundreds of proteins. Major proteins sumoylated by PIAS1 E3 SUMO Ligase involves nucleic acid binding proteins, transcription factors, and cytosolic proteins which is interesting to note. It is also interesting to note that the SUMO sites on the intermediate filament proteins keratin and vimentin are highly conserved in different species.

By itself, the study provides a crucial and novel mechanistic finding that PIAS1-mediated SUMOylation of vimentin is required for the incorporation of soluble tetramers into VIFs. Furthermore, the really intriguing aspect of this study is that this could provide an explanation why overexpression of vimentin is a hallmark of many aggressive and metastasizing cancer cells. Moreover, the sumo site detection protocol is strongly verified as shown by NSMCE2 and PFDN2 sumo sites along with vimentin.

The paper is well written in simple, understandable language and most of the claims are supported by solid experimental data. The approach adopted to supply a global view of PIAS1 substrates and then focus on vimentin as a potential target, (partly due to increased recent interest) was well founded and resulted in some intriguing results.

The model at the end gives a good summary of the data on how the interplay of PTMs such as phosphorylation and sumoylation on vimentin affects filament dynamics.

Overall, this is an important addition to the knowledge of the physiological roles PIAS1 and demonstrated the potential of PIAS1 as a sumoylation agent directed toward cytoskeletal proteins. The connection to intermediate filament is both surprising and intriguing and has broad ramifications both in normal conditions as well as in cancers and potentially also in other disease in which vimentin is critically involved (for example, wound healing and fibrosis)

Specific points

Fig 1) PIAS1 Overexpression Promotes cell Proliferation and Motility

Fig 1c does not look convincing. It would be important to get better images with sufficient resolution to clearly see the individual cells . Same applies to Fig 1e for Boyden chamber assay images

Fig 5) PIAS1 SUMOylation Promotes its Localization to PML:NBs. As indicated by the immunofluorescence studies, approximately 45% of WT PIAS1-GFP colocalized with PML (Fig. 5b). However, only 10 cells are used for this analysis, which comes across as a low number. From the images provided, there seems to be colocalization taking place. This argument would be significantly strengthened by a quantitative approach, for example, by using Pearson's coefficient and Mander's coefficient, both of which are commonly used for colocalization analysis. Analysis of larger cell population would also be warranted.

No significant changes in the colocalization of PIAS1 and PML were observed when experiments were repeated using either single or double PIAS1-GFP mutants (data not shown). This information would be useful to show in the supplementary data

It is stated that there is a 50% reduction in PIAS1-GFP-PML colocalization when all three sites were mutated, suggesting a possible functional redundancy among these sites (Figs. 5b and 5c). The provided images indicated even less or no colocalization at all. This aspect should be strengthened experimentally and also explained better.

The fact that the co-localization of PML and PIAS1-GFP was not totally abrogated for the triple mutant might be explained by residual interactions between SUMOylated PML and the SIM of PIAS1-GFP32. This would need more extensive elaboration to be intelligible.

Reviewer #3 (Remarks to the Author):

The manuscript by Li et al, describes a proteomic approach aimed at identification of PIAS1 substrates using quantitative mass spectrometry. Overall, the manuscript is nicely written and the focus of the manuscript addresses a challenging task within the SUMOylation field, i.e. the identification of direct substrates of SUMO E3 ligases. However, the manuscript does not entail any biological novelty or conceptual advance, while the technical quality of the manuscript and the supporting work is not of a quality that would warrant publication in Nature Communications.

The main concern related to the cell system chosen, where overexpression of both SUMO and PIAS1 does not seem properly controlled, which is going to cause artefacts – an effect that the authors indirectly admit to by observing a lot of increased cytoplasmic SUMOylation. Thus the system will give rise to many pleiotropic effects. This could have been fundamentally addressed, e.g. by using an inducible overexpression (for equal and low levels), using an opposed control experiment (for example a knockdown), or inclusion of multiple PIAS proteins (to figure out redundancy).

Likewise, the SILAC setup is somehow chosen in a weird manner... the authors use 3 channels and of these 2 are used for the exact same PIAS overexpression! Why not just use two channels then, or use the third for something sensible?

Moreover, the entire validation of targets is done in vitro and in vivo, with the in vivo meaning in cells and with overexpression of all SUMO components. Thus meaning it's also just in vitro.

Detailed comments:

- Regarding the levels of PIAS1 overexpression, the error bars in Fig. 1A are not defined. How many replicates were performed? The displayed WB looks overexposed, meaning the difference in expression could easily be larger than 2.5-fold.

- Transient overexpression is usually not homogenous, and only a fraction of the cells will get transfected at considerably different levels (several orders of magnitude). The WB indicates an overall increase of 2.5-fold, but the differences from cell to cell are likely to be more dramatic. To get more equal levels of overexpression, a stable cell line should be made with an inducible construct that allows a more equal induction of overexpression.

- Along the same lines, transient overexpression for 48 h with PIAS1 is likely to result in all manners of pleiotropic effects down the line, since many cellular processes will be affected. Thus, it is impossible to know whether any observed effects are direct or indirect biological consequences. To address this, similarly as above, a stable cell line should be created where overexpression of PIAS1 can be performed for a more reasonable period of time (e.g. 6 to 24 hours).

- The levels of PIAS1 overexpression for the proteomics experiments, as displayed in Fig. 2B, are vastly beyond what was shown in Fig. 1A. The difference in expression here could easily be 50 to 100-fold. How does this make sense compared to the WB shown in Fig. 1A, which is essentially the same experiment, which shows a 2.5-fold with an extremely small standard deviation that would imply this overexpression is reproducible?

- The proteomic method is essentially identical to what the authors have previously published, and should thus not be portrayed as innovative, especially since the number of identifications are also lower than what the authors previously reported.

- The experimental setup is somewhat confusing. Why were 2 of 3 channels dedicated to the exact same Myc-PIAS1 vector? The experiment could have been better controlled by using the third channel for either a different expression level of PIAS1, for example by using a viral transduction or a stable cell line, or alternatively a knockdown of PIAS1 could have been performed.

- For the selection of "putative" sites and substrates, as displayed in Fig. 3B, what sort of statistical testing was performed? The authors only state a p-value of less than 0.05, but there does not appear to be a correction for multiple hypothesis testing or another form of permutation-based FDR? Combined with the fact that many SUMO targets would be upregulated because of large PIAS1 overexpression, this could result in overestimation of the number of regulated targets.

- The authors observe an increase in cytosolic proteins being SUMOylated when PIAS1 is overexpressed, and indicate this may be because of nucleocytoplasmic shuttling. Rather, this could be related to huge overexpression of PIAS1 which will very likely result in mislocalization of PIAS1, and as a result, mislocalization of the SUMOylation process itself. The authors should perform subcellular localization analysis using microscopy to confirm that the overexpressed PIAS1 is properly localized in the nucleus of the cells.

- The combination of PIAS1 overexpression and the cell line used, in which epitope-tagged and mutant SUMO is also overexpressed, may exponentially lead to pleiotropic effects and SUMOylation of proteins which would not be SUMOylated under physiological conditions.

- In a sense, the authors highlight this problem indirectly by demonstrating that another SUMO E3 ligase (NSMCE2) is increasingly SUMOylated by PIAS1. This could be an example of pleiotropic effect, as it is impossible to discern whether NSMCE2 is directly modified by PIAS1, or rather NSMCE2 auto-modifies itself because of increased activity. The in vitro control experiments performed are hardly direct evidence since they in no way mimic the cellular environment in which SUMOylation usually takes place.

- Similarly, in the confirmatory in vivo experiments, the authors overexpress PIAS1, the target protein, and use the cell line with SUMO already overexpressed. With so many components of the reaction overexpressed into the cell, this can hardly be called in vivo and is very likely to not be physiological (and probably should also be called in vitro). The authors should confirm putative PIAS1 targets in a fashion that is better controlled. Ideally, there should be no overexpression of SUMO in the cell lines, and no overexpression of the proteins to be verified (endogenous antibodies should be used).

- As an alternative to overexpression of PIAS1, the authors should use knockdown of PIAS1 to show the inverse effect – a decreased SUMOylation of the putative targets. The knocked-down pool of (endogenous) PIAS1 could then reasonably be expected to be at proper expression levels and localization, and should thus have an effect on the SUMOylation levels of the targets.

- In the co-localization experiment, Fig. 5B, there are numerous PIAS1 bodies that do not co-localize with the relatively few PML bodies. What exactly are all these nuclear bodies where PIAS1 does reside? It appears that only a minor fraction of PIAS1 co-localizes with PML even under the best conditions. How does endogenously expressed PIAS1 localize in the cell? This is a control that should be included to ensure that the observed localization of PIAS1 is not merely an artefact of overexpression.

- The SUMOylation of VIM as shown in Fig. 6b is entirely unclear – there is no labeling for the WB, and only a tiny cropped image of a single band is shown, even though the authors mention two sites.

- Moreover, the lack of SUMOylation of VIM in the 2KR mutant may not necessarily stem from the two mutations, but rather from the fact that two mutations were made in highly conserved residues? Such mutations could directly alter the structure, function, or localization of VIM. If the localization of the mutant VIM would be altered, this could affect its ability to be SUMOylated, and the authors should verify subcellular localization of the WT and mutant VIM.

- The authors further make an unfounded claim with Fig. 7A, stating that SUMOylation of VIM “clearly” benefits its solubility. Rather, as outlined above, if the two mutations in VIM directly compromise the localization and stability of VIM, this would be the reason why the protein never gets SUMOylated, and why it is otherwise not active in the biological functions tested. This is a classical chicken-or-the-egg scenario, and the authors do not have any evidence to support their claim.

- Since the two sites mutated are at the very C-terminus of VIM, the authors should be able to rescue the 2KR mutant of VIM by C-terminally fusing two copies of SUMO. This is just one potential control experiment the authors could include to substantiate their claims.

- The SUMOylation of VIM has been previously reported by the authors (Lamoliatte et al., 2017), with the exact same two sites. In the context of the current work, this would have made some sense if PIAS1 were to be considered, but in Figures 6 and 7 there is no inclusion of PIAS1 in any of the experiments. This makes the manuscript rather disjointed.

- At the end of the results section, the authors link PIAS1 back to regulation of VIM, even though (as mentioned above) there are no actual direct experiments performed with PIAS1 and VIM. This makes the main conclusion entirely speculative and based on correlative evidence, rather than mechanistic and backed up by causative evidence.

Response to review of Manuscript "Quantitative proteomics identifies novel PIAS1 protein substrates involved in cell migration and motility" - **NCOMMS-18-36228**

Italic highlights response on specific comments raised by the reviewers

Bold highlights places where modifications have been made in the revised manuscript, also highlighted in yellow in the revised manuscript

We thank all reviewers for their time, constructive comments and feedback on our work. Please find a point-by-point response to each comment below.

Reviewer #1 (Remarks to the Author):

Protein Inhibitor of Activated STAT 1 (PIAS1), with or without its SUMO ligase function, has been implicated in several important cellular pathways. There is also interesting evidence showing that the PIAS1 is overexpressed in several human cancers. Thibault's group has here used quantitative proteomics to identify PIAS1's targets in a proteome-wide fashion in HEK293 cells. This approach led to identification of novel, putative PIAS1's SUMO ligase substrates, including vimentin, involved in cell migration and motility. This is an interesting work utilizing state-of-the-art SUMO proteomics approaches.

1) However, the actual role of PIAS1's SUMO ligase function in the promotion of cell proliferation and mobility remains unclear and requires clarification. To strengthen the message of this manuscript, a SUMO ligase function-mutated/dead version of PIAS1 (e.g. SP-RING 2xCys→Ser point mutant) should be tested in parallel with wtPIAS1 in HeLa cell-based proliferation, cell migration and invasion assays of experiments Figure 1 and Figure 6.

We fully agree with the reviewer's comment and feel that this information is indispensable for the message we want to convey. As a result, we made the C351S variant of PIAS1, which was supposed to abolish the activity of the E3 ligase^{1,2}. Unfortunately, a residual activity appeared to remain since the SUMOylation of VIM was still observed with the PIAS1 variant.

In the event that another variant of PIAS1 may still give residual activity and to be 100% sure that there was no PIAS1 activity in the cell we decided to make a CRISPR-CAS9 mediated PIAS1 knockout cell line and performed the proliferation and migration assays again.

We updated Figure 1 (see image above) and edited the text on p. 6 to reflect the new results using the PIAS1 KO cells.

To investigate the physiological function of PIAS1 in HeLa cells, we overexpressed PIAS1 (Fig. 1a) and generated a PIAS1 knockout cell line (Fig. 1b). For the PIAS1 overexpression, the abundance of PIAS1 in HeLa cells was increased by 6-fold at 48 h post-transfection (Fig. 1a). PIAS1 overexpression promotes HeLa cell proliferation by ~50% (Fig. 1c), while the knockout cell reduced the rate of proliferation by ~50% (Fig. 1d). We further examined the phenotypic effects of PIAS1 expression on cell migration using the wound-healing assay. The migration ability of HeLa cells was increased after

PIAS1 overexpression (Fig. 1e), while the knockout cell displayed a reduced rate of migration. (Fig. 1g). Taken together, these results highlight the role that PIAS1 plays in regulating cell growth and cell migration of HeLa cells.

Reviewer #2 (Remarks to the Author):

The article provides a novel quantitative approach for SUMO proteome analysis and identified new SUMO sites on hundreds of proteins. Major proteins sumoylated by PIAS1 E3 SUMO Ligase involves nucleic acid binding proteins, transcription factors, and cytosolic proteins which is interesting to note. It is also interesting to note that the SUMO sites on the intermediate filament proteins keratin and vimentin are highly conserved in different species.

By itself, the study provides a crucial and novel mechanistic finding that PIAS1-mediated SUMOylation of vimentin is required for the incorporation of soluble tetramers into VIFs. Furthermore, the really intriguing aspect of this study is that this could provide an explanation of why overexpression of vimentin is a hallmark of many aggressive and metastasizing cancer cells. Moreover, the sumo site detection protocol is strongly verified as shown by NSMCE2 and PFDN2 sumo sites along with vimentin.

The paper is well written in simple, understandable language and most of the claims are supported by solid experimental data. The approach adopted to supply a global view of PIAS1 substrates and then focus on vimentin as a potential target, (partly due to increased recent interest) was well founded and resulted in some intriguing results.

The model at the end gives a good summary of the data on how the interplay of PTMs such as phosphorylation and sumoylation on vimentin affects filament dynamics.

Overall, this is an important addition to the knowledge of the physiological roles PIAS1 and demonstrated the potential of PIAS1 as a sumoylation agent directed toward cytoskeletal proteins. The connection to intermediate filament is both surprising and intriguing and has broad ramifications both in normal conditions as well as in cancers and potentially also in other disease in which vimentin is critically involved (for example, wound healing and fibrosis)

Specific points

1) Fig 1) PIAS1 Overexpression Promotes cell Proliferation and Motility. Fig 1c does not look convincing. It would be important to get better images with sufficient resolution to clearly see the individual cells. Same applies to Fig 1e for Boyden chamber assay images

We repeated the cell migration and provided improved images for Figures 1e, 1f and Figure 6c (wound healing assay) and Figure 6. Ultimately, we opted to remove the invasion assay to create continuity

between Figure 1 and Figure 6. We felt that the number of Boyden chamber assays required for figure 6 was too overwhelming considering the limited time needed to revise our manuscript.

2) Fig 5) PIAS1 SUMOylation Promotes its Localization to PML:NBs. As indicated by the immunofluorescence studies, approximately 45% of WT PIAS1-GFP colocalized with PML (Fig. 5b). However, only 10 cells are used for this analysis, which comes across as a low number. From the images provided, there seems to be colocalization taking place. This argument would be significantly strengthened by a quantitative approach, for example, by using Pearson's coefficient and Mander's coefficient, both of which are commonly used for colocalization analysis. Analysis of larger cell population would also be warranted.

We thank the reviewer for this insightful comment. By performing the analysis with more cells the data was of much better quality. This improved technical quality highlighted a statistical significance between the K315R variant of PIAS1 and WT form. The message is much clearer than before.

Barplot representation of the (a) Mander's coefficient and (b) Pearson's coefficient for the colocalization of each PIAS1 construct and PML (** $p < 0.01$, Student's t-test, $n = 20$ cells/condition).

We modified the text on p. 11 as follows to include our new findings:

Of all the single variants tested, a significant change in the colocalization of PIAS1 and PML was observed only with the K315R alteration (Fig. 5c). However, we noted a greater than 50% reduction in PIAS1-GFP–PML colocalization when all three sites were mutated, suggesting a possible cooperativity among these sites (Figs. 5b and 5c).

3) No significant changes in the colocalization of PIAS1 and PML were observed when experiments were repeated using either single or double PIAS1-GFP mutants (data not shown). This information would be useful to show in the supplementary data.

Our above observation is now obsolete with the new data we gathered. We did however include the new data for the single mutant PML-GFP colocalization in Figures 5b, 5c.

4) It is stated that there is a 50% reduction in PIAS1-GFP-PML colocalization when all three sites were mutated, suggesting a possible functional redundancy among these sites (Figs. 5b and 5c). The provided images indicated even less or no colocalization at all. This aspect should be strengthened experimentally and also explained better.

We performed the colocalization experiments with 20 cells to improve the statistics of the analysis. The HEK293 cells we used are mildly adherent and only a few cells remain on the microscope slides after fixing, washing and staining, which greatly hinders our ability to quantify large populations on cells. We also added the following text on p. 11 to the manuscript to explain why we believe several SUMOylation sites on PIAS1 are needed for its proper PML localization.

This functional redundancy may be required to ensure the proper localization of PIAS1 to PML nuclear bodies under different biological context. Also, the cooperative nature of multiple SUMOylation events to enhance affinity has been noted before, where the affinity of RNF4 for SUMO dimers is 10-fold higher than for the monomer³³.

5) The fact that the co-localization of PML and PIAS1-GFP was not totally abrogated for the triple mutant might be explained by residual interactions between SUMOylated PML and the SIM of PIAS1-GFP32. This would need more extensive elaboration to intelligible.

We agree with the reviewer that the above statement was not fully explained properly. To address this we added the following text on p. 11 the manuscript.

Indeed, we have shown that several sites on PIAS1 are SUMOylated, which aids to localize PIAS1 to the PML bodies. However, PML itself is also SUMOylated on several lysine residues. Therefore non-SUMOylated PIAS1 can still localize, albeit less efficiently, to PML nuclear bodies via the SIM that is located on PIAS1 and the SUMOylated moieties on PML. A similar phenomenon was reported by our group for the SUMO E2 protein UBC9³⁴.

Reviewer #3 (Remarks to the Author):

The manuscript by Li et al, describes a proteomic approach aimed at identification of PIAS1 substrated using quantitative mass spectrometry. Overall, the manuscript is nicely written and the focus of the manuscript addresses a challenging task within the SUMOylation field, i.e. the identification of direct substrates of SUMO E3 ligases. However, the manuscript does not entail any biological novelty or conceptual advance, while the technical quality of the manuscript and the supporting work is not of a quality that would warrant publication in Nature Communications.

The main concern related to the cell system chose, where overexpression of both SUMO and PIAS1 does not seem properly controlled, which is going to cause artefacts – an effect that the authors indirectly admit to by observing a lot of increased cytoplasmic SUMOylation. Thus the system will give rise to many pleiotropic effects. This could have been fundamentally addressed, e.g. by using an inducible overexpression (for equal and low levels), using an opposed control experiment (for example a knockdown), or inclusion of multiple PIAS proteins (to figure out redundancy).

Likewise, the SILAC setup is somehow chosen in a weird manner... the authors use 3 channels and of these 2 are used for the exact same PIAS overexpression! Why not just use two channels then, or use the third for something sensible?

Moreover, the entire validation of targets is done in vitro and in vivo, with the in vivo meaning in cells and with overexpression of all SUMO components. Thus meaning it's also just in vitro.

Detailed comments:

1) Regarding the levels of PIAS1 overexpression, the error bars in Fig. 1A are not defined. How many replicates were performed? The displayed WB looks overexposed, meaning the difference in expression could easily be larger than 2.5-fold.

In accordance with the reviewer's comment we quantified the level of PIAS1 overexpression using a less exposed western blot membrane with 5 biological replicates. The reviewer's comment was justified since the shorter exposure revealed that the PIAS1 fold change was 6-fold rather than the reported 2.5-fold. As a result, the fold change reported in the manuscript was corrected on p. 6.

2) Transient overexpression is usually not homogenous, and only a fraction of the cells will get transfected at considerably different levels (several orders of magnitude). The WB indicates an overall increase of 2.5-fold, but the differences from cell to cell are likely to be more dramatic. To get more equal levels of overexpression, a stable cell line should be made with an inducible construct that allows a more equal induction of overexpression.

We do agree with this reviewer's comment. Although stable cell lines with an inducible vector have their benefits, we feel as though this method is best for this project. Indeed, the DOX addition (small molecule generally used to induce protein expression) may skew the biological results since this drug is known to adversely affect cell proliferation as well as metabolism^{3,4}. Also, we believe we performed the necessary controls to rule out phenotypic artifacts as indicated below.

a) We validate the phenotypic effects we observed with PIAS1 overexpression with the knockout cell line. We found an inverse phenotype with the knockout cells, meaning that what was observed phenotypically with the PIAS1 overexpression was genuine. These new findings are shown in Figures 1d and 1f. Accordingly, we observed slower cell migration (Figure 1f) and proliferation (Figure 1d) with the PIAS1 KO system.

b) As for the VIM overexpression, we found once again a contrasting phenotype for the mutant VIM expressing cells that can be partially rescued with the addition of PIAS1 (Figure 6d). We added the following text to the manuscript on p. 12 to reflect this new finding.

However, VIM^{mt} alone conferred no effect on cell migration, while transfecting PIAS1 with VIM^{mt} rescued the phenotypic effect (Figs. 6c-d).

c) Lastly, the proteomic workflow was conducted with several biological replicates (including reverse SILAC labelling to eliminate variations in SILAC labelling efficiencies) to mitigate biological artifacts that may have come from PIAS1 overexpression.

3) Along the same lines, transient overexpression for 48 h with PIAS1 is likely to result in all manners of pleiotropic effects down the line, since many cellular processes will be affected. Thus, it is impossible to know whether any observed effects are direct or indirect biological consequences. To address this, similarly as above, a stable cell line should be created where overexpression of PIAS1 can be performed for a more reasonable period of time (e.g. 6 to 24 hours).

We originally chose the 48 h time point for the proteomic analysis since the level of PIAS expression was greatest. See image below, which shows the higher level of PIAS1 expression overall as well as increased proportion of expressing cells. Importantly, as shown in figure 3a, the proteome does not change after the PIAS1 overexpression indicating that globally, transcriptional events remain unaffected. Also, the effects observed on cell proliferation and migration were reproducible across all biological replicates obtained. If the observed pleiotropic effects were artifacts as suggested by the reviewer, these would not be consistently observed resulting in low p-values across biological replicates. Since the effects observed showed high biological reproducibility, we surmised that these are biologically relevant pathways with meaningful information to be gained. It is noteworthy that SUMOylation is known to be highly dynamic, where protein SUMOylation occurs in less than 30 min during a heat shock suggesting that a 6 h induction would still lead to pleiotropic effects. Finally, several important studies have used similar transient overexpression systems to gain further biological insights into ubiquitin and SUMO signaling

pathways^{5, 6, 7, 8}. We believe that the use of PIAS1 overexpression is legitimate to understand the role of this important mediator of the SUMO pathway.

4) The levels of PIAS1 overexpression for the proteomics experiments, as displayed in Fig. 2B, are vastly beyond what was shown in Fig. 1A. The difference in expression here could easily be 50 to 100-fold. How does this make sense compared to the WB shown in Fig. 1A, which is essentially the same experiment, which shows a 2.5-fold with an extremely small standard deviation that would imply this overexpression is reproducible?

We would like to clarify that the phenotyping was done with HeLa cells and the proteomic experiments with HEK293 cells. This is because HEK293 cells (as well as the HEK293-SUMO3 cells that were generated for our previous proteomic studies) do not adhere properly to microscope slides and hence are not good systems for microscopy and cell migration assays.

After reevaluations, and in accordance with point 1, the PIAS1 fold change in the HeLa cells was slightly higher with a fold change above 6 between control and PIAS1 overexpressing cells. The fold-change in the HEK293 cells was of 20 for the PIAS1 level between control and PIAS1 overexpressing cells with a p-value of 0.00016, meaning that within each cell line, the biological reproducibility was excellent.

5) The proteomic method is essentially identical to what the authors have previously published, and should thus not be portrayed as innovative, especially since the number of identifications are also lower than what the authors previously reported.

We will comply with the comment and have downplayed the novelty of the method by referring to the method as an expansion on our previous workflow rather than it being novel. We would like to note however that the number of sites was lower than our previous studies because the cells were not stimulated with MG132 nor heat shock, which is routinely used in large scale SUMO proteomic studies to increase the number of sites. For this project however, we avoided these stimuli to limit the number of variables.

6) The experimental setup is somewhat confusing. Why were 2 of 3 channels dedicated to the exact same Myc-PIAS1 vector? The experiment could have been better controlled by using the third channel for either a different expression level of PIAS1, for example by using a viral transduction or a stable cell line, or alternatively a knockdown of PIAS1 could have been performed.

We opted to perform this SILAC labelling strategy to increase the number of biological replicates to garner higher quality data (better p-values). Due to the multistep purification process more technical variability arises. Indeed, the Ni-NTA purification step and peptide level IP both produce sample variability. It is also due to this technical variability that SILAC labelling was used. The labelling allowed to reduce the inter sample variability between control and 2 PIAS1 channels by pooling the SILAC channels together prior to sample processing, but could not control variability between the different pooled SILAC samples that were run in parallel. To increase the statistical confidence of the SUMO site fold changes we opted to use 6-biological replicates.

Also, including another transfection level as one of the SILAC channels would have reduced the number of biological replicates to N=3. We found in our previous projects that SUMO proteomics data tend to have more missing values (SUMO sites that are not quantified in certain replicates) than typical large scale proteomic studies (again, this is due to the multistep workflow). For this reason, having only 3 replicates will lead to a considerable loss of quantified SUMO site since SUMO site must be quantified in at least 3 replicates to obtain a meaningful p-value.

7) For the selection of “putative” sites and substrates, as displayed in Fig. 3B, what sort of statistical testing was performed? The authors only state a p-value of less than 0.05, but there does not appear to be a correction for multiple hypothesis testing or another form of permutation-based FDR? Combined with the fact that many SUMO targets would be upregulated because of large PIAS1 overexpression, this could result in overestimation of the number of regulated targets.

We did not originally correct for multiple hypothesis testing based on the known substrates of PIAS1, which included PCNA SUMOylation. When correcting the analysis we lost this known PIAS1 target. But, we do agree with the reviewer that we should be more stringent with the analysis to mitigate the number of false targets. Therefore we will perform the correction to increase the quality of the output.

Of note, in an ideal situation, it should be expected that no SUMOylation sites be reduced in abundance as a result of the PIAS1 overexpression. Under this assumption, which is the best case scenario, then the new corrected data also has a <5% FDR. Indeed, 4 sites out of the 95 regulated sites are down regulated, corresponding to <5%. Again, this is assuming no sites are down regulated by PIAS1 overexpression. If, on the other hand, some sites are down regulated by the overexpression due to some degradation process for example, then the FDR is well below the 5% threshold. We also updated all panels in Figures 3 and 4, together with Supplementary Figures 4, 6, 7 with the new corrected cutoff.

8) The authors observe an increase in cytosolic proteins being SUMOylated when PIAS1 is overexpressed, and indicate this may be because of nucleocytoplasmic shuffling. Rather, this could be related to huge overexpression of PIAS1 which will very likely result in mislocalization of PIAS1, and as a result, mislocalization of the SUMOylation process itself. The authors should perform subcellular localization analysis using microscopy to confirm that the overexpressed PIAS1 is properly localized in the nucleus of the cells.

The more stringent Benjamini-Hochberg correction reduced significantly the number of regulated sites including those associated with cytoplasmic proteins. Correspondingly, cytoplasmic proteins are no longer a regulated GO term in Figure 3e. Furthermore, we also performed IF studies with anti-PIAS1 to look at the endogenous protein, the GFP signal for a PIAS1-GFP construct and anti-myc for the myc-PIAS1 protein, which was used in the proteomic experiments. The results below confirm that the PIAS1 protein, under all conditions (low and high expression), is localized virtually exclusively to the nuclei in a PML speckle organization, just like the endogenous protein.

9) The combination of PIAS1 overexpression and the cell line used, in which epitope-tagged and mutant SUMO is also overexpressed, may exponentially lead to pleiotropic effects and SUMOylation of proteins which would not be SUMOylated under physiological conditions.

The use of an exogenous SUMO construct to determine and quantify protein SUMOylation under different cell stimulation paradigms has been described in several publications. We studied the level of expression of the SUMO3 construct and it's in vitro conjugation exhaustively^{9, 10, 11, 12}. Of note, we used the same stably expressing clone for this study as we did for the previous studies and transfected PIAS1 in the same cell line. Briefly, in vitro conjugation assays showed that the SUMO3 variant conjugates with the same efficiency as the wild-type SUMO3⁹. Moreover, the level of overexpression of the SUMO3 variant is approximately 2-3 fold greater than that of the endogenous SUMO3.

As for the PIAS1 overexpression, we used the requantify function in Maxquant, which looks for the SILAC triplets in the background to report a fold-change. This means that SUMO3 artifacts created by the PIAS1 overexpression (that would therefore be absent in the control condition) would return extremely large fold-changes that would be greater than 10-fold. This would show up on the volcano plot of Figure 3b as showing extremely large fold changes with poor p-values, which was not observed. As for the extent of PIAS1 overexpression that could lead to pleiotropic effects, we compared the overlap between sites identified in this study and those from recent large scale SUMO proteome analyses¹¹, and found that more than 90% of sites were common. This comparison supports the notion that protein SUMOylation following PIAS1 overexpression targets sites previously reported in large-scale studies and does not result in unanticipated identification.

10) In a sense, the authors highlight this problem indirectly by demonstrating that another SUMO E3 ligase (NSMCE2) is increasingly SUMOylation by PIAS1. This could be an example of pleiotropic effect, as it is impossible to discern whether NSMCE2 is directly modified by PIAS1, or rather NSMCE2 auto-modifies itself because of increased activity. The in vitro control experiments performed are hardly direct evidence since they in no way mimic the cellular environment in which SUMOylation usually takes place.

We feel that the answer to point 3) addresses this comment as well. We added the following text on p. 17 of the manuscript discussion to acknowledge the potential role of pleiotropic effects.

However, due to the versatile activity of PIAS1 in the cell it is possible that some of the SUMOylation sites that increased upon PIAS1 expression are not direct substrates of PIAS1 E3 ligase activity but may in fact stem from other pleiotropic effects that are E3 ligase independent³⁶. This includes the regulation of transcription factors such as the inhibition of STAT1 by protein sequestration¹⁹.

11) Similarly, in the confirmatory in vivo experiments, the authors overexpress PIAS1, the target protein, and use the cell line with SUMO already overexpressed. With so many components of the reaction overexpressed into the cell, this can hardly be called in vivo and is very likely to not be physiological (and probably should also be called in vitro). The authors should confirm putative PIAS1 targets in a fashion that is better controlled. Ideally, there should be no overexpression of SUMO in the cell lines, and no overexpression of the proteins to be verified (endogenous antibodies should be used).

At the reviewers request we changed the title from in vivo SUMOylation assay to in vitro cell based SUMOylation assay in the text and removed all reference to the in vivo aspect.

The identification of PIAS1 substrates without overexpression would be significantly challenging due to the low proportion of modified residues and the lack of appropriate tools to detect substrates in a site-specific manner. Usually a SUMO enrichment needs to be performed from hundreds of 100 ug size amounts to be detected by WB. We attempted variations to our protocol such as not transfecting the Flag substrate in the cell and performed the same workflow (ie: only transfect PIAS1 in the SUMO3m cells) and could not have enough SUMOylated substrate for a signal on WB. It should be noted that we started from 8 mg of cell extract for the proteomic studies while only 20 ug of protein can be loaded on a SDS-PAGE gel, which is why we can identify these sites by MS but cannot observe the SUMOylated pattern by WB under non-stimulated conditions. As can be observed in the Rank N plot below, the PIAS1

substrates that were analyzed in this study (orange circles) are of much lower abundance than proteins previously observed as SUMOylated using WB (grey circles).

Performing such assays by transfecting the SUMO conjugation machinery is routine and some groups have even transfected the E2 protein as well in order to observe the SUMOylation of their substrate by WB^{13, 14, 15}. Although this method is not optimal, it is the only method that allow the study of low abundance SUMOylation events in the cell by WB.

12) As an alternative to overexpression of PIAS1, the authors should use knockdown of PIAS1 to show the inverse effect – a decreased SUMOylation of the putative targets. The knocked-down pool of (endogenous) PIAS1 could then reasonably be expected to be at proper expression levels and localization, and should thus have an effect on the SUMOylation levels of the targets.

Reducing the abundance of PIAS1 is not the optimal approach due to the documented redundancy amongst PIAS family members^{14,16,17}. Although reducing PIAS1 levels by up to 80% by shRNA or siRNA might seem considerable, the effects on specific substrates can be minimal. Indeed, the four PIAS family members show a considerable degree of substrate overlap. Therefore, knocking down PIAS1 levels by 5-fold may not reduce the substrate SUMOylation to the same magnitude. If another PIAS family member is also responsible for the SUMOylation of the same substrate then the observed fold change is now less than 2-fold. To obtain high-quality data that is reproducible and with good p-value the fold-change must be significant as well. Obtaining good p-value with low fold-change differences is not easily achievable with low abundant species, like SUMOylated peptides. On the other hand, If we overexpress PIAS1 by 20-fold (MS data), the SUMOylated substrate will increase more considerably leading to larger changes and increased p-values. As indicated for Reviewer 1 point1, we did perform PIAS1 KO experiments and confirmed reciprocal phenotypes to PIAS1 overexpression. We modified the text on p. 5 accordingly:

The migration ability of HeLa cells was increased after PIAS1 overexpression (Fig. 1e), while the knockout cell displayed a reduced rate of migration. (Fig. 1g). Taken together, these results highlight the role that PIAS1 plays in regulating cell growth and cell migration of HeLa cells.

13) In the co-localization experiment, Fig. 5B, there are numerous PIAS1 bodies that do not co-localize with the relatively few PML bodies. What exactly are all these nuclear bodies where PIAS1 does reside? It appears that only a minor fraction of PIAS1 co-localizes with PML even under the best conditions. How does endogenously expressed PIAS1 localize in the cell? This is a control that should be included to ensure that the observed localization of PIAS1 is not merely an artifact of overexpression.

Please see point 8) above for the IF with endogenous PIAS1. Clearly, endogenous PIAS1 also localizes in nuclear speckles that look like those seen with Myc-PIAS1 in Fig. 5B. We also looked at the localization of GFP-PIAS1 and found the same phenomenon. We cannot identify the nature of those PIAS1 nuclear speckles, other than it does not rely on the SIM of PIAS1 as documented by Brown et al ¹⁸. It should be noted that the exact same pattern was observed in 4 different cell lines by Brown et al. (Figure 1 of their article).

14) The SUMOylation of VIM as shown in Fig. 6b is entirely unclear – there is no labeling for the WB, and only a tiny cropped image of a single band is shown, even though the authors mention two sites.

To improve the quality of the results we performed the assay again with and without PIAS1 (see image below, now Figure 6b in revised manuscript). We did not crop the image this time and show the SUMOylation smear of VIM^{wt}. Importantly, the SUMOylation of VIM^{wt} is increased when PIAS1 is added and little SUMOylation is observed for the VIM^{mut}.

15) Moreover, the lack of SUMOylation of VIM in the 2KR mutant may not necessarily stem from the two mutations, but rather from the fact that two mutations were made in highly conserved residues? Such mutations could directly alter the structure, function, or localization of VIM. If the localization of

the mutant VIM would be altered, this could affect its ability to be SUMOylated, and the authors should verify subcellular localization of the WT and mutant VIM.

Unfortunately, the K to R variations are the closest alterations possible to maintain the charge of the amino side chain. We looked at the localization of the variant and wild-type VIM in Figure 7b and found that both proteins localized in the same structures, except with varying proportions. In addition, we found with follow-up experiments that upon transfecting PIAS1 (see image in point 14) more of the VIM^{mt} protein is soluble and partly SUMOylated. These results mean that if PIAS1 is added there is some slight level of SUMOylation that occurs on the VIM^{mt}, which helps solubilize a greater fraction of it. This is also evidenced on the cell motility results (see image below) that show that adding VIM^{mt} and PIAS1 to HeLa cells provides a greater motility than PIAS1 alone, suggesting that VIM^{mt} is indeed functional but is hampered by its poor potential to be SUMOylated (requires PIAS1 overexpression to be sufficiently SUMOylated). Taken together, these results indicate that the VIM^{mt} protein is functional.

Figure 6. SUMOylation of Vimentin (VIM) is required for proper cell growth, migration and invasion in HeLa cells. (a) VIM sequence indicating SUMOylation sites at Lys 439 and Lys 445 localized in the tail domain, and highly conserved across six different species. (b) Western blot analysis of HeLa cells transfected with an empty vector as a negative control, Wild-type vimentin (Flag-VIM^{wt}), Vimentin K439, 445R, double mutant (Flag-VIM^{mt}), with or without Myc-PIAS1. (c) VIM^{mt} expression inhibits proper cell migration and can be rescued by PIAS1 expression, as determined by a wound-healing assay. (d) Quantification of the wound healing assay for biological triplicates. (*p<0.05, **p<0.01, Student's t-test).

16) The authors further make an unfounded claim with Fig. 7A, stating that SUMOylation of VIM “clearly” benefits its solubility. Rather, as outlined above, if the two mutations in VIM directly compromise the localization and stability of VIM, this would be the reason why the protein never gets SUMOylated, and why it is otherwise not active in the biological functions tested. This is a classical chicken-or-the-egg scenario, and the authors do not have any evidence to support their claim.

Please see point 15 above.

17) Since the two sites mutated are at the very C-terminus of VIM, the authors should be able to rescue the 2KR mutant of VIM by C-terminally fusing two copies of SUMO. This is just one potential control experiment the authors could include to substantiate their claims.

This is an interesting and insightful idea. However, fusing copies of SUMO on the C-terminus of the protein could not recover the phenotype since the SUMOylation and deSUMOylation are dynamic. That is, the VIM polymerization dynamics rely on both conjugation and deconjugation. In the same vein, we did however transfect PIAS1 along with VIM^{mt}, which slightly SUMOylated VIM^{mt} and recovered the migration phenotype (see image for point 15).

18) The SUMOylation of VIM has been previously reported by the authors (Lamoliatte et al., 2017), with the exact same two sites. In the context of the current work, this would have made some sense if PIAS1 were to be considered, but in Figures 6 and 7 there is no inclusion of PIAS1 in any of the experiments. This makes the manuscript rather disjointed.

We agree with the reviewer’s comment and updated Figures 6 to include PIAS1 transfections along with VIM. Check point 15.

19) At the end of the results section, the authors link PIAS1 back to regulation of VIM, even though (as mentioned above) there are no actual direct experiments performed with PIAS1 and VIM. This makes the main conclusion entirely speculative and based on correlative evidence, rather than mechanistic and backed up by causative evidence.

We validated our claims with new experiments shown in Figure 6 and 7 of the main text, where the addition of PIAS1 promotes both the solubility of VIM^{mt} and also rescues the VIM^{mt} phenotype (see point 15 above) linking PIAS1 to VIM related phenotypes.

References (response to reviewers)

- 1 Lee, J. M. *et al.* PIAS1 enhances SUMO-1 modification and the transactivation activity of the major immediate-early IE2 protein of human cytomegalovirus. *FEBS letters* **555**, 322-328 (2003).
- 2 Sudharsan, R. & Azuma, Y. The SUMO ligase PIAS1 regulates UV-induced apoptosis by recruiting Daxx to SUMOylated foci. *J Cell Sci* **125**, 5819-5829, doi:10.1242/jcs.110825 (2012).
- 3 Ahler, E. *et al.* Doxycycline alters metabolism and proliferation of human cell lines. *PLoS One* **8**, e64561, doi:10.1371/journal.pone.0064561 (2013).
- 4 Chatzisprou, I. A., Held, N. M., Mouchiroud, L., Auwerx, J. & Houtkooper, R. H. Tetracycline antibiotics impair mitochondrial function and its experimental use confounds research. *Cancer Res* **75**, 4446-4449, doi:10.1158/0008-5472.CAN-15-1626 (2015).

- 5 Totland, M. Z. *et al.* The E3 ubiquitin ligase NEDD4 induces endocytosis and lysosomal sorting of connexin 43 to promote loss of gap junctions. *J Cell Sci* **130**, 2867-2882, doi:10.1242/jcs.202408 (2017).
- 6 Cajanek, L., Glatter, T. & Nigg, E. A. The E3 ubiquitin ligase Mib1 regulates Plk4 and centriole biogenesis. *J Cell Sci* **128**, 1674-1682, doi:10.1242/jcs.166496 (2015).
- 7 Potts, P. R. & Yu, H. Human MMS21/NSE2 is a SUMO ligase required for DNA repair. *Mol Cell Biol* **25**, 7021-7032, doi:10.1128/MCB.25.16.7021-7032.2005 (2005).
- 8 Bischof, O. *et al.* The E3 SUMO ligase PIASy is a regulator of cellular senescence and apoptosis. *Mol Cell* **22**, 783-794, doi:10.1016/j.molcel.2006.05.016 (2006).
- 9 Galisson, F. *et al.* A novel proteomics approach to identify SUMOylated proteins and their modification sites in human cells. *Molecular & cellular proteomics : MCP* **10**, M110 004796, doi:10.1074/mcp.M110.004796 (2011).
- 10 Lamoliatte, F. *et al.* Large-scale analysis of lysine SUMOylation by SUMO remnant immunoaffinity profiling. *Nat Commun* **5**, 5409, doi:10.1038/ncomms6409 (2014).
- 11 McManus, F. P., Lamoliatte, F. & Thibault, P. Identification of cross talk between SUMOylation and ubiquitylation using a sequential peptide immunopurification approach. *Nat Protoc* **12**, 2342-2358, doi:10.1038/nprot.2017.105 (2017).
- 12 Lamoliatte, F., McManus, F. P., Maarifi, G., Chelbi-Alix, M. K. & Thibault, P. Uncovering the SUMOylation and ubiquitylation crosstalk in human cells using sequential peptide immunopurification. *Nat Commun* **8**, 14109, doi:10.1038/ncomms14109 (2017).
- 13 Hsieh, Y. L. *et al.* Ubc9 acetylation modulates distinct SUMO target modification and hypoxia response. *EMBO J* **32**, 791-804, doi:10.1038/emboj.2013.5 (2013).
- 14 Uzoma, I. *et al.* Global Identification of Small Ubiquitin-related Modifier (SUMO) Substrates Reveals Crosstalk between SUMOylation and Phosphorylation Promotes Cell Migration. *Molecular & cellular proteomics : MCP* **17**, 871-888, doi:10.1074/mcp.RA117.000014 (2018).
- 15 Yan, Y. L. *et al.* DPPA2/4 and SUMO E3 ligase PIAS4 opposingly regulate zygotic transcriptional program. *PLoS Biol* **17**, e3000324, doi:10.1371/journal.pbio.3000324 (2019).
- 16 Palvimo, J. J. PIAS proteins as regulators of small ubiquitin-related modifier (SUMO) modifications and transcription. *Biochem Soc Trans* **35**, 1405-1408, doi:10.1042/BST0351405 (2007).
- 17 Tahk, S. *et al.* Control of specificity and magnitude of NF-kappa B and STAT1-mediated gene activation through PIASy and PIAS1 cooperation. *Proc Natl Acad Sci U S A* **104**, 11643-11648, doi:10.1073/pnas.0701877104 (2007).
- 18 Brown, J. R. *et al.* SUMO Ligase Protein Inhibitor of Activated STAT1 (PIAS1) Is a Constituent Promyelocytic Leukemia Nuclear Body Protein That Contributes to the Intrinsic Antiviral Immune Response to Herpes Simplex Virus 1. *J Virol* **90**, 5939-5952, doi:10.1128/JVI.00426-16 (2016).

Reviewers' comments:

Reviewer #1 (Remarks to the Author):

The authors have responded to my main points in a satisfactory manner.

Reviewer #2 (Remarks to the Author):

This is an important study that has the ingredients of being a hallmark study that will open up entirely new avenues with broad ramifications for research related to vimentin and its functions in cells. It also has several features that will be of key interest for SUMO-focused researchers as well as researchers working on SUMO-related PTM-proteomics. Publication of this study will, therefore, be interesting for and serve a broad community of researchers.

The study includes a broad range of data, very well presented, and of high quality. The text is easy to follow with very logical conclusions. The conclusions are concise and accurate without being too far-reaching or far-fetched.

The authors have done an excellent job in the revision. The revised version has exhaustively addressed all issues raised by this reviewer but, as far as I can see, also the issues raised by the two other reviewers.

Reviewer #3 (Remarks to the Author):

In the revised manuscript, the authors have provided elaborate feedback on the previous reviewer comments. Most of these do relate to textual comments while experimental changes are very minimal.

The main concern over the manuscript still resides with the overall accuracy of the presented proteomics data, in particularly with regard to readers of the journal who might want to follow up on identified candidates in the presented screen.

The overarching problem still relate to the cell system used, where the authors overexpress both PIAS and SUMO, which inherently will lead to pleiotropic effects, and consequently result in the identification of SUMO proteins that would not be SUMOylated under physiological conditions.

Following this notion, the authors themselves provide in their point-by-point response a concerning comment which further raises doubt over the cell system. In the initial review, the authors were asked to perform the experiment upon knockout of PIAS1 - which the authors discard by stating that PIAS enzymes exhibit a documented redundancy, which renders knockout experiments impracticable.

This statement furthers the concern regarding overexpression of PIAS (and SUMO), as the redundancy between the PIAS enzymes will exist both when the enzymes are reduced or increased in expression. So when overexpressing one PIAS, the overall redundancy will lead to the pleiotropic effects exactly as raised by the reviewer, where proteins primarily are being modified via PIAS1 due to redundancy rather than due to specificity.

So following the authors own note, they will have to demonstrate that redundancy among the PIAS systems is not a problem when overexpression the enzymes.

Response to review of Manuscript "Quantitative proteomics identifies novel PIAS1 protein substrates involved in cell migration and motility" - **NCOMMS-18-36228A-Z**

Italic highlights response on specific comments raised by the reviewers

Bold highlights places where modifications have been made in the revised manuscript, also highlighted in yellow in the revised manuscript

We thank all reviewers for their time, constructive comments and feedback on our work. Please find a point-by-point response to each comment below.

Reviewer #1 (Remarks to the Author):

The authors have responded to my main points in a satisfactory manner.

We thank the reviewer for his/her comments and for valuable suggestions to improve our manuscript.

Reviewer #2 (Remarks to the Author):

This is an important study that has the ingredients of being a hallmark study that will open up entirely new avenues with broad ramifications for research related to vimentin and its functions in cells. It also has several features that will be of key interest for SUMO-focused researchers as well as researchers working on SUMO-related PTM-proteomics. Publication of this study will, therefore, be interesting for and serve a broad community of researchers.

The study includes a broad range of data, very well presented, and of high quality. The text is easy to follow with very logical conclusions. The conclusions are concise and accurate without being too far-reaching or far-fetched.

The authors have done an excellent job in the revision. The revised version has exhaustively addressed all issues raised by this reviewer but, as far as I can see, also the issues raised by the two other reviewers.

We are most grateful for the constructive criticisms of this reviewer and for commenting on the significance of our contribution.

Reviewer #3 (Remarks to the Author):

The main concern over the manuscript still resides with the overall accuracy of the presented proteomics data, in particular with regard to readers of the journal who might want to follow up on identified candidates in the presented screen. The overarching problem still relate to the cell system used, where the authors overexpress both PIAS and SUMO, which inherently will lead to pleiotropic effects, and consequently result in the identification of SUMO proteins that would not be SUMOylated under physiological conditions. Following this notion, the authors themselves provide in their point-by-point response a concerning comment which further raises doubt over the cell system. In the initial review, the authors were asked to perform the experiment upon knockout of PIAS1 - which the authors discard by stating that PIAS enzymes exhibit a documented redundancy, which renders knockout experiments impracticable. This statement furthers the concern regarding overexpression of PIAS (and

SUMO), as the redundancy between the PIAS enzymes will exist both when the enzymes are reduced or increased in expression. So when overexpressing one PIAS, the overall redundancy will lead to the pleiotropic effects exactly as raised by the reviewer, where proteins primarily are being modified via PIAS1 due to redundancy rather than due to specificity. So following the authors own note, they will have to demonstrate that redundancy among the PIAS systems is not a problem when overexpression the enzymes.

We thank the reviewer for this comment which relates to point 12 of the previous review regarding the use of knockdown of PIAS1 to show the inverse effect and the concern on the potential redundancy between PIAS E3 ligases. To address this point, we used CRISPR/Cas9 gene editing technology with sgRNA specific to each of the four PIAS E3 ligases (e.g. PIAS1, PIAS2, PIAS3 and PIAS4) and a scrambled sgRNA for the transfection in HEK293 SUMO3m cells. We cultured KO and control HEK293 SUMO3m cells in triplicate using light, medium, and heavy SILAC media. A portion of each cell extract was used to confirm the KO of individual PIAS by western blotting as indicated below.

We then combined light, medium and heavy cells so that control (CTL) cells along with two different PIAS KO cells were present in each SILAC triplet for a total of 6 samples (e.g. Rep1-6) as indicated in the scheme below.

SUMOylated proteins from each sample were enriched on Ni-NTA beads while the flow through eluate was kept separately for proteome quantitation and normalization. Trypsin was used to digest SUMO proteins on Ni-NTA beads and flow through proteins. We used SUMO remnant immunopurification to isolate SUMOylated peptides prior to LC-MS/MS analysis of the corresponding digests on a Q-Exactive HF mass spectrometer. Targeted LC-MS/MS analyses were performed with an inclusion list to acquired MS/MS spectra of the $[M+3H]^{3+}$ precursor ions of the isotopically labeled SUMOylated peptides ETNLDLPLVDTHSK*R and TLLIK*TVETR where * indicates SUMOylation site. The targeted MS data were analyzed using Skyline and fragment ions for each targeted mass were extracted, and peak areas were integrated. We also analyzed by LC-MS/MS in data-dependent acquisition the tryptic peptides from the flow through proteins to normalize protein abundance across the 6 different samples as illustrated below.

Confirmation of the SUMOylated vimentin peptides was achieved for each sample, and an example is shown below for Rep 3.

We then determined the fold change ratio of SUMOylated peptides between each PIAS KO and control condition after normalizing SILAC mixing ratios based on the protein abundance between samples. Results obtained are depicted in the bar plots below (n=3).

These LC-MS/MS analyses confirmed that PIAS1 is the only E3 SUMO ligase that imparted a significant reduction of vimentin SUMOylation upon CRISPR/Cas9 gene knock out. With the exception of PIAS4 that showed an increase in vimentin SUMOylation none of the other PIAS KO affected vimentin SUMOylation. The increase in vimentin SUMOylation imparted by PIAS4 KO might be explained by an increase in SUMO protease activity. These experiments along with those shown in Figure 6 and in the accompanying text clearly indicate that PIAS1 targets specifically the SUMOylation of vimentin at K439 and K445. Accordingly, we added the following text on p.12 and 18 of our revised manuscript along with supplementary figures 9-10:

On p. 12:

To confirm that VIM is selectively SUMOylated by PIAS1, we used CRISPR/Cas9 gene editing technology with sgRNA specific to each of the four PIAS E3 ligases (e.g. PIAS1, PIAS2, PIAS3 and PIAS4) and a scrambled sgRNA for the transfection in HEK293 SUMO3m cells. KO and control HEK293 SUMO3m cells were cultured in triplicate using light, medium, and heavy SILAC media (Fig. S9). Following Ni-NTA enrichment, SUMOylated proteins were digested on beads with trypsin and modified tryptic peptides were isolated by SUMO remnant immunoaffinity purification prior to targeted LC-MS/MS analyses using an inclusion list to detect and identify individual isotopically labeled SUMOylated peptides of VIM (e.g. ETNLDSLPLVDTHSK*R and TLLIK*TVETR where * indicates SUMOylation site). MS/MS spectra of isotopically-labeled tryptic peptides were used to confirm identification of SUMOylated VIM peptides (Fig. S10). We also analyzed by LC-MS/MS in data-dependent acquisition the tryptic peptides from the flow through proteins to normalize protein abundance across the 6 different samples. These quantitative proteomics experiments revealed that PIAS1 selectively targeted the SUMOylation of VIM at K439 and K445.

On p. 18:

We uncovered that PIAS1 specifically SUMOylates K439 and K445 residues of VIM. This modification increased the solubility of VIM and is correlated with the uptake of ULF onto VIF in a phospho-dependent mechanism.

We also modified the on line methods section to include additional experiments:

In Cell Culture and Transfection

PIAS1-sgRNA sequence: TTCTGAACTCCAAGTACTGT, PIAS2-sgRNA sequence: CAAGTATTACTAGGCTTTGC, PIAS3-sgRNA sequence: GCCCTTCTATGAAGTCTATG, PIAS4-sgRNA sequence: GGCTTCGCGCCGTAGTCTTAG and scrambled sgRNA sequence: GGCTTCGCGCCGTAGTCTTA.

SILAC labeling and extraction

Similar conditions were used for the culture of PIAS KO cells using the combination scheme described in Fig. S9.

Mass Spectrometry Analyses

For targeted LC-MS/MS analyses of PIAS KO cells generated through CRISPR/Cas9, SUMOylated peptides obtained from SUMO remnant immunoaffinity enrichment were analyzed using an inclusion list to detect and identify each isotopologues of the VIM SUMOylated peptides at K439 (e.g. ETNLDSLPLVDTHSK*R) and K445 (e.g. TLLIK*TVETR). The SUMO peptides were analyzed by LC-MS/MS on an Orbitrap Q Exactive HF system using a 120min gradient, similar to the proteomics measurements described above. The mass spectrometer was operated in a targeted-MS2 acquisition mode with a maximum injection time of 1000 ms, 1 microscan, 30 000 resolution, 2E5 AGC target, 1.6 m/z isolation window, and 25% normalized collision energy.

Proteins from the flowthrough of the Ni-NTA purification step were trypsinized and analyzed to normalize protein amounts across sample sets. The data-dependent acquisition method used was similar to the proteomics measurements described above.

Data Processing

The flow through proteome MS data were searched with PEAKS X engine (Bioinformatics Solutions, Inc.) against the UniProt/SwissProt database (<http://www.uniprot.org/>) released on June 05, 2019.

The precursor tolerance was set to 10 ppm and fragment ion tolerance to 0.01Da. The maximum missed cleavage sites for trypsin was set to 2. Carbamidomethylation (C) was set as a fixed modification and oxidation (M), deamination (NQ) and NQTGG (K), $^2\text{H}_4$ -lysine and $^{13}\text{C}_6$ -arginine (SILAC medium) and $^{15}\text{N}_2^{13}\text{C}_6$ -lysine and $^{15}\text{N}_4^{13}\text{C}_6$ -arginine (SILAC heavy) were set as variable modifications with a maximum of five modifications per peptide. The false discovery rate for peptides was set to 1.0% with decoy removal. Proteins were quantified with ≥ 2 unique peptides. The relative change in protein abundance across samples were determined using the PEAKS X software. The targeted MS data were analyzed using Skyline version 19.1, MacCoss Lab Software, Seattle, WA; (<https://skyline.ms/wiki/home/software/Skyline/page.view?name=default>), fragment ions for each targeted mass were extracted, and peak areas were integrated. Fold change ratios between the CTL and KO samples were calculated based on peak areas after normalizing peak intensities using the normalization factor determined from the proteome analysis.

REVIEWERS' COMMENTS:

Reviewer #3 (Remarks to the Author):

The authors have adequately addressed the concerns of the reviewer.